

# Temperature and mineral dust variability recorded in two low accumulation Alpine ice cores over the last millennium

Pascal Bohleber[1,2,3], Tobias Erhardt[4,5], Nicole Spaulding[1], Helene Hoffmann[2], Hubertus Fischer[4,5], and Paul Mayewski[1]

[1]Climate Change Institute, University of Maine, Orono, Maine, USA
[2]Institute of Environmental Physics, Heidelberg University, Heidelberg, Germany
[3]Institute for Interdisciplinary Mountain Research, Austrian Academy of Sciences, Innsbruck, Austria
[4]Climate and Environmental Physics, Physics Institute, University of Bern, Bern, Switzerland
[5]Oeschger Centre for Climate Change Research, University of Bern, Bern, Switzerland

*Correspondence to:* Pascal Bohleber (Pascal.Bohleber@iup.uni-heidelberg.de)

**Abstract.** Among ice core drilling sites in the European Alps, the Colle Gnifetti (CG) glacier saddle is the only one to offer climate records back to at least 1000 years. This unique long-term archive is the result of an exceptionally low net accumulation driven by wind erosion and rapid annual layer thinning. To-date, however, the full exploitation of the CG time series has been hampered by considerable dating uncertainties and the seasonal summer bias in snow preservation. Using a new core drilled

in 2013 we extend annual layer counting, for the first time at CG, over the last 1000 years and add additional constraints to the resulting age scale from radiocarbon dating. Based on this improved age scale, and using a multi-core approach with a neighboring ice core, we explore the potential for reconstructing long-term temperature variability from the stable water isotope and mineral dust proxy time series. A high and potentially non-stationary isotope/temperature sensitivity limits the quantitative use of the stable isotope variability thus far. However, we find substantial agreement comparing the mineral dust proxy $Ca^{2+}$

with instrumental temperature. The temperature-related variability in the $Ca^{2+}$ record is explained based on the temperature-dependent snow preservation bias combined with the advection of dust-rich air masses coinciding with warm temperatures. We show that using the $Ca^{2+}$ trends for a quantitative temperature reconstruction results in good agreement with instrumental temperature and the latest summer temperature reconstruction derived from other archives covering the last 1000 years. This includes a "Little Ice Age" cold period as well as a medieval climate anomaly. In particular, part of the medieval climate period

around 1100–1200 AD stands out through an increased occurrence of dust events, potentially resulting from a relative increase in meridional flow and dry conditions over the Mediterranean.

## 1 Introduction

Glaciers and ice caps of high mountain ranges can provide climate records of mid- and low latitudes complementary to polar ice cores. In comparison to their polar counterparts, mountain drilling sites are characterized by a comparatively small-scale

glacier geometry and their proximity to continental source areas. As a consequence, cold mountain glaciers are an especially worthwhile target for ice core studies focusing on Holocene climate, e.g. in view of the envisaged IPICS 2k array (Brook et al.,



2006) and the present underrepresentation of ice core records contributing to the PAGES 2k Network (Ahmed et al., 2013). In the European Alps, ice core studies have been performed at Col du Dôme, Mont Blanc (Preunkert et al., 2000), Fiescherhorn, Bernese Alps (Schwerzmann et al., 2006), Ortles, Eastern Alps (Gabrielli et al., 2016) as well as at Colle Gnifetti and Colle del Lys in the Monte Rosa region (e.g. Wagenbach et al., 2012, and references therein). Among these glaciers, Colle Gnifetti (CG)

– in spite of its limited glacier depth – stands out as the only site where net snow accumulation is low enough to provide records over the last millennium and potentially beyond at a reasonable time resolution. The exceptionally low net accumulation at CG is a result of seasonal net snow loss by wind erosion: Since snow consolidation is most effective during the summer half year, winter precipitation is more likely to be removed from the surface (Wagenbach, 1992). This has far-reaching consequences with respect to the interpretation of the CG ice cores, hampering to-date the full exploitation of their unique long climate time

series. On the one hand, considerable uncertainty in the individual ice core chronologies becomes an obstacle already after a few hundred years. Difficulties in deploying annual layer counting as the main dating tool arise from snow scouring, rapid layer thinning associated with strongly non-linear time-depth relationships and the extremely low time resolution achieved in the bottom part of the glacier by conventional cm-resolution analyses. As a consequence, dating the deeper part of CG ice cores is commonly based on simple extrapolation combined with constraints from radiocarbon analysis (e.g. Jenk et al., 2009).

On the other hand, irregular and summer-biased snow deposition makes the annual or long-term levels of ice core proxy signals with a prominent seasonal cycle a primary function of the relative winter snow fraction preserved, as opposed to their common climatological meaning. In addition, net snow accumulation is characterized by substantial spatial and temporal variability, leading to considerable influence of upstream flow effects and depositional noise (Wagenbach, 1992). In contrast to the strong signals of anthropogenic aerosol increase, depositional noise especially challenges the detection of the comparatively weak

stable water isotope trends ($\delta^{18}$O and $\delta$D). Under these circumstances, the comparison of multiple cores drilled at the same site can be used to identify an atmospheric signal as shared variability among the cores (Bohleber et al., 2013; Wagenbach et al., 2012).

Here we present new results to tackle the two-fold challenge above with a new core drilled at Colle Gnifetti in 2013. In order to obtain a reliable long-term chronology, we utilize state-of-the-art continuous flow analysis for ice core impurity profiling

and, to identify even highly thinned annual layers, laser ablation inductively coupled plasma mass spectrometry (LA-ICP-MS) at sub-mm depth resolution (Della Lunga et al., 2017; More et al., 2017; Haines et al., 2016; Mayewski et al., 2014). We combine annual layer counting in the resulting impurity profiles with absolute age constraints from radiocarbon analysis, taking advantage of recent progress in applying this technique to mountain ice cores (Uglietti et al., 2016; Hoffmann, 2016). Based on a refined long-term chronology, the time series of stable water isotopes and mineral dust proxies are investigated for

long-term temperature variability. For this purpose we integrate datasets from an additional ice core drilled in 2005 on the same flow line as the 2013 core.



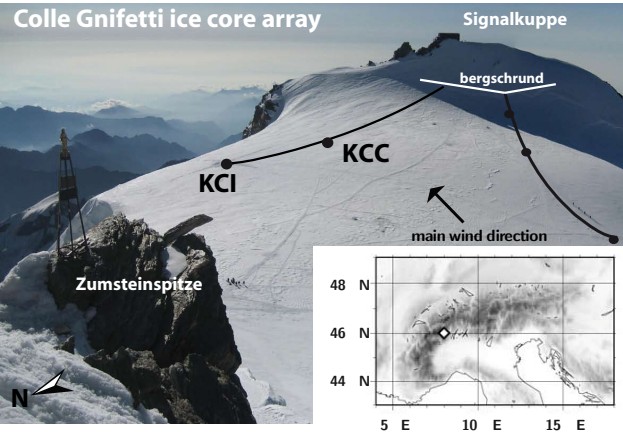

**Figure 1.** The ice core array at Colle Gnifetti, at 4450 m asl in the Monte Rosa summit range. The drilling sites of the two cores KCI and KCC are located on approximately the same flow line (black line) towards the eastern flank (downwind of the main wind direction), hence providing their same upstream catchment area. Locations of previous drillings initiated by the Institute of Environmental Physics are also shown as small black dots for reference.

## 2  Glaciological settings of the CG drilling site

Details on the glaciological features of CG are described thoroughly in the literature; e.g. Haeberli et al. (1988); Lüthi and Funk (2000); Konrad et al. (2013) for geometry and glacier flow, Haeberli and Funk (1991); Hoelzle et al. (2011) for englacial temperature and Alean et al. (1983) for surface accumulation. Here, we present only a brief overview, mainly dedicated to explaining the role of snow deposition in relation to recording atmospheric temperature and mineral dust variability in the CG ice cores.

### 2.1  Snow deposition influence

With a horizontal scale of 400 m and a maximum ice thickness of around 140 m, the CG site forms a small firn saddle at around 4500 m asl between two summits of the Monte Rosa massif. The orientation of the convex, central saddle axis coincides with the main westerly wind direction, thereby making the downwind-situated ice cliff a perfect sink for drifting snow (Figure 1). Hence, a substantial fraction of the annual fresh snow precipitation is removed at CG, which prevents any link of the climatologic precipitation rate to the net snow accumulation rate. The latter ranges from 0.15 m water equivalent (WE) per year in the north-facing flank to about 1.2 m WE per year in the southern one, where the higher abundance of ice layers and ice crusts significantly reduces the snow erosion rate (Alean et al., 1983). Within the CG north flank (comprising our CG ice core array) fresh snow consolidation is faster during the summer half year (additionally supported by refreezing surface melt). Accordingly, the mean net snow accumulation is mainly made up by precipitation of the warm seasons, which entails a systematic over-representation of the summer half-year in chemical/isotopic conditions (Wagenbach, 1989). As a





consequence, the isotope/impurity co-variation on the inter-annual scale reflects to a large degree changes in the amount of winter precipitation contributing to annual mean values (Wagenbach, 1992).

Saharan dust deposition events, a frequent phenomenon in the Alps with main occurrence in spring and summer (Prodi and Fea, 1979), yield an additional short-term effect on snow deposition at CG. A single deposition event typically lasts less than a few
days (Sodemann et al., 2005; Schwikowski et al., 1995). The associated warm air temperature and the substantially lowered snow albedo both support surface snow consolidation and partly protect the dust layer from wind erosion. Intensive Saharan dust events of the summer half year, associated with directly northward transport of air masses, are most likely to become preserved at CG. Saharan dust layers in CG ice cores can be characterized by high concentrations of insoluble particles, $SO_4^{2-}$ and $Ca^{2+}$ coinciding with buffered low acidity, as well as to some extent by increased $\delta^{18}O$ and deuterium excess values.
Therefore, the $Ca^{2+}$ record of the CG ice cores is primarily related to mineral dust and dominated by Saharan dust related spikes (Wagenbach et al., 1996; Wagenbach and Geis, 1989).

## 2.2   The role of snow deposition in a possible $Ca^{2+}$-temperature relationship

A long-term co-variation between $Ca^{2+}$ and $\delta^{18}O$ suggesting a possible relationship between climate and dust deposition at CG has already been noted in earlier studies but was left for future investigation (Wagenbach and Geis, 1989; Wagenbach
et al., 1996). Evaluating the various temporal cycles contained in the $Ca^{2+}$ and $\delta^{18}O$ records requires separating the (local) snow preservation changes from the (remote) atmospheric signals. A later study specifically targeted at establishing the link between the $\delta^{18}O$ signal and air temperature changes in the presence of the snow preservation influence used for this purpose a multi-core comparison against an instrumental temperature dataset adjusted to reflect the summer bias at CG. This comparison revealed not only a shared isotope variability among the decadal scale trends of the CG cores but also a dominant influence
of atmospheric temperature on shared isotope variability (Bohleber et al., 2013). Considering the co-variation of the long-term variability of i) $Ca^{2+}$ and $\delta^{18}O$, and ii) $\delta^{18}O$ and temperature, strongly motivates expecting atmospheric temperature variability also reflected in the $Ca^{2+}$ signal. Worth noting in this context, temperature-related variability has also been observed for other impurities, e.g. for $NH_4^+$ at a low-latitude site (Kellerhals et al., 2010a), albeit explained by a different mechanism than discussed for $Ca^{2+}$ at CG here. A potential driver for a $Ca^{2+}$-temperature relationship can be expected from i) the advec-
tion of air masses comprising a high dust load generally being associated with warm temperatures, ii) the deposition of dust leading to lowered snow albedo thus supporting surface snow consolidation and iii) temperature affecting snow preservation. For instance, warm summers feature increased vertical mixing and hence a higher atmospheric impurity load, and in addition, entail faster fresh snow consolidation. This may lead to an increased relative amount of impurity-rich summer snow deposition. Accordingly, snow preservation generally plays an important role not only with respect to coupling of seasonal varying signals,
but also for introducing a temperature-related imprint to impurity time series.

While it remains difficult to quantify the influence of the above contributions i) and ii), we made an attempt to semi-quantitatively explore the imprint of snow preservation on the long-term variability in $Ca^{2+}$, using $\delta^{18}O$ as a reference. Previous studies used simplified conceptual models to investigate the influence of snow deposition on seasonal ice core signals, and demonstrated the decisive role played by the amplitude of the seasonality (Wagenbach et al., 2012; Fisher and Koerner, 1988). All major





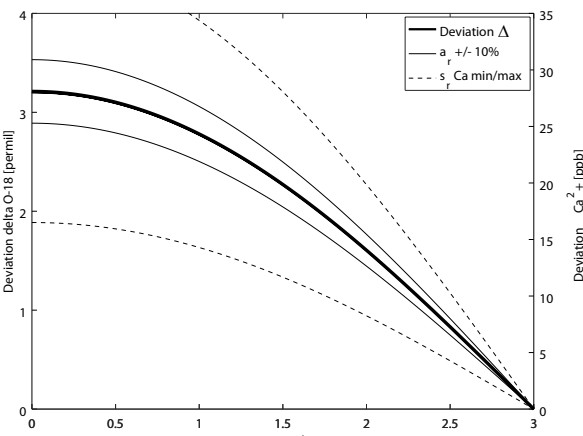

**Figure 2.** Sensitivity of the average $Ca^{2+}$ signal to snow preservation effects, in comparison to $\delta^{18}O$. The plot shows the deviation of the mean annual ice core signal from the respective precipitation mean as a function of phase shift t between the seasonal cycles of the signal and the accumulation rate. The thick solid black line shows the deviation for $\delta^{18}O$ (from Wagenbach et al. (2012)) and $Ca^{2+}$, plotted on the y-axes on the left and right axis, respectively, and using typical seasonality parameters at Colle Gnifetti (see text). Indicated as thin lines are ranges corresponding to a variability of the relative seasonal amplitude $a_r$ of $10\%$ (arbitarily chosen), and using the estimated maximum uncertainty range of $s_r$ for $Ca^{2+}$ (solid and dashed lines, respectively).

impurity species at CG (e.g. including $NH_4^+$ and $Ca^{2+}$) show a distinct seasonality together with an increasing trend over the twentieth century (e.g. Döscher et al., 1996; Wagenbach et al., 1996). $NH_4^+$ shows the strongest summer-winter contrast of roughly a factor of 9, while the mean summer to winter ratio at CG remains close to 5 for $Ca^{2+}$. Due to the episodic input of Saharan dust, $Ca^{2+}$ shows a more flickered seasonal cycle and hence a larger uncertainty in determining representative summer

and winter levels (Preunkert et al., 2000). Notably, although a $Ca^{2+}$-seasonality was also found at sites close to CG, this feature appeared absent at drilling sites on the northern side of the Alps (Eichler et al., 2004).

Here we follow the model of Wagenbach et al. (2012), which assumes sinusoidal cycles for the precipitation-borne signal $S(t)$ and the surface accumulation pattern $A(t)$, and a phase-lag $t_\phi$ between $S(t)$ and $A(t)$. The deviation $\Delta$ of the mean signal recorded in the firn at CG ($\bar{S}_{\text{firn}}$) with respect to the overall mean signal ($S_0$) in the precipitation then becomes (corresponding

to equation (5) in Wagenbach et al. (2012)):

$$S(t) = S_0(1 + s_r \sin\omega(t + t_\phi)) \qquad\qquad A(t) = A_0(1 + a_r \sin\omega t) \qquad\qquad \Delta = \bar{S}_{\text{firn}} - S_0 = S_0\,s_r\,a_r\,\cos\omega t_\phi \qquad (1)$$

with $a_r$ and $s_r$ denoting the relative signal amplitudes, $t_\phi$ the temporal phase shift between the two cycles and $\omega$ the cycle frequency equal to $2\pi$/T (period T = 1 yr). We reproduced here the calculation for $\delta^{18}O$ by Wagenbach et al. (2012) with typical seasonality parameters at Colle Gnifetti (e.g. $a_r = 0.8$). To estimate $\Delta$ for $Ca^{2+}$, we used values reported by Preunkert et al.

(2000) for CG, i.e. taking $S_0 = 112$ ppb (the average of typical summer and winter concentrations) as well as $s_r = 0.64 \pm 0.26$. Notably, this value of $s_r$ for $Ca^{2+}$ is close to the absolute value of $s_r = 0.5$ obtained for $\delta^{18}O$ by Wagenbach et al. (2012).



Estimating the radiation-induced control of snow consolidation on the seasonal net accumulation cycle at CG is consistent with a phase shift of $t_\phi = 1.5$ months (broadly equal to the delay of the isotope/aerosol peak with respect to insolation (Wagenbach et al., 2012)). Figure 2 shows the deviation $\Delta$ for $Ca^{2+}$ and $\delta^{18}O$ as a function of phase shift $t_\phi$ and $a_r$.

The framework of the conceptual model allows us to semi-quantitatively explore the temperature-snow preservation coupling

on the long-term $Ca^{2+}$ signal. For instance, the effect of warmer atmospheric summer temperatures on snow preservation is envisaged as increased summer snow deposition, corresponding to an according change in $a_r$ and, an additional possibility, a change in phase-shift $t_\phi$. Evidently, a decrease in phase-shift $t_\phi$ also increases the depositional effect, which is maximal once both seasonal cycles are fully in-phase ($t_\phi = 0$). We highlight the sensitivity in the deviation corresponding to an arbitrarily chosen variability of $a_r$ of $\pm 10\%$ and $t_\phi = 1.5 \pm 0.5$ months. This already shifts the mean $Ca^{2+}$ level by about 15 ppb, which is

in the same order of magnitude as the long-term trends of around $\pm 50$ ppb found in previous studies (Wagenbach et al., 1996) and also in the core investigated here. Accordingly, it seems reasonable to assume that a systematic influence of temperature on snow preservation (contribution iii) above) plays a important role with respect to driving the $Ca^{2+}$-temperature co-variation. The potential variability in the $Ca^{2+}$-seasonality due to the episodic input of Saharan dust can add a substantial contribution to the deviation (cf. dashed lines in Figure 2). However, the influence of a single Saharan dust event is rather short-term in

comparison to the envisaged systematic shifts to $a_r$ and $t_\phi$ imposed by atmospheric temperature change. In this context, long-term trends in the occurrence rate of Saharan dust events becomes a feature of interest. Changes in the dust occurrence rate can originate from changes in the meridional versus zonal circulation and/or in the desert dust source strength. In this case detecting the frequency of dust peaks in the ice core matters, which is less affected by depositional variability than the dust level itself.

## 3   Ice core analysis

The two cores used in this study, denoted as KCI and KCC, were drilled in 2005 and 2013, respectively. Both cores were drilled roughly on the same flow line, making them the natural choice for our inter-core comparison, i.e. opposed to using previously deep cores drilled on another flow line (Figure 1). Table 1 summarizes basic glaciological parameters of the two cores. The depth sections used in this study were chosen to comprise roughly the last 1000 years, i.e., the upper 44 m WE (corresponding

to $81\%$ relative depth) and 35 m WE ($73\%$ relative depth) of KCC and KCI, respectively. Table 2 provides an overview of the carefully co-registered datasets used in this study. The various methods of analysis are discussed briefly in the following.

### 3.1   Impurity profiles from continuous flow analysis

Continuous flow analysis (CFA) of the KCC core was performed with the setup at the Division for Climate and Environmental Physics, Physics Institute, at the University of Bern. Analyses performed on the meltwater flow included meltwater conductiv-

ity, insoluble particle concentration and size distribution as well as selected ion species ($Ca^{2+}$, $NH_4^+$, $NO_3^-$, $Na^+$, see Table 2). In addition, stable water isotopes were analyzed using a Picarro instrument coupled directly to the meltwater flow. The size distribution of insoluble particles recorded by the optical particle sensor was used to derive a profile of the "coarse particle



**Table 1.** Basic glaciological parameters of the two CG ice cores

| Core name | KCI | KCC |
|---|---|---|
| Position GPS (WGS84) | N 45.92972 E 7.87696 | N 45.92893 E 7.87627 |
| Year of drilling | 2005 | 2013 |
| Total depth [m abs] | 61.84 | 71.81 |
| Total depth [m WE] | 48.44 | 53.77 |
| Surface net accumulation [cm WE/yr] | 14 | 22 |
| Firn-ice-transition [m WE] | 17 | 21 |

percentage" (CPP). The CPP was calculated based on particle volume, and represents the percentage of particles exceeding a threshold of 4.0 $\mu$m. The threshold was chosen such that it corresponds to the expected median particle diameter of Saharan dust particles at CG (Wagenbach and Geis, 1989). Deviations from a CPP of 50% indicate higher or lower contribution of large and small particles respectively. The melt rate was adjusted to provide the necessary amount of water for all analyses

resulting in an effective depth resolution ranging from 1.2 cm at the very top of the core to about 0.5 cm for all depth below approximately 25 m WE. Electrical conductivity measurements (ECM) performed at the Institute of Environmental Physics, Heidelberg University were used primarily to obtain a qualitative record of the acidity of the ice in connection to the detection of Saharan dust events.

The KCI core was analyzed using the reduced CFA setup at the Institute of Environmental Physics, Heidelberg University.

Meltwater conductivity and insoluble particle concentration were measured by CFA at about 0.7 cm effective resolution. Continuous sub-sampling of the core for stable water isotope analyses was conducted at a depth resolution typically ranging between 5 and 10 cm. Due to the relatively high firn temperature at CG, isotope smoothing is much faster compared to polar sites with similar annual layer thickness. Hence re-sampling most of KCI even at 1.5 cm depth resolution did not significantly restore any high-frequency isotope variability (Bohleber et al., 2013).

**3.2 Ultra-high resolution Ca-profile of the KCC core by laser ablation ICP-MS**

Laser ablation inductively coupled plasma mass spectrometry (LA-ICP-MS) was conducted in the WM Keck Laser Ice Facility at the Climate Change Institute (University of Maine) and used to analyze $^{44}$Ca at ultra-high depth resolution (better than 120 $\mu$m). The more abundant $^{40}$Ca is blocked by mass interference from $^{40}$Ar used as carrier gas. Details regarding the method, sample preparation and calibration routine can be found in Sneed et al. (2015). Briefly, the components of this system

include a Thermo Element 2 ICP-MS, a New Wave UP-213 laser, and a cryo-cell chamber, designed to seal a 1 m ice core from the surrounding air while maintaining a uniform temperature of $-15°$ C. In order to ensure a complete seal of the ablation chamber, porous firn parts could not be measured. From 29.5 m WE to bedrock, the KCC ice core was analyzed for $^{44}$Ca along a single ablation track. Crucial for further deployment for annual layer counting, the trend components in the LA-ICP-MS



**Table 2.** Overview on ice core analyses and datasets used in this study

| Core | Parameters | Sampling | Effective resolution [cm] |
|------|-----------|----------|---------------------------|
| **KCC** | Meltwater Conductivity, $NH_4^+$, $NO_3^-$, $Na^+$ | Continuous Flow | > 0.5 |
| | Insoluble Particles, $Ca^{2+}$ | Continuous Flow | > 0.5 |
| | Stable water isotopes ($\delta^{18}O$ and $\delta D$) | Continuous Flow | > 0.5 |
| | Electric Conductivity | ECM | > 0.5 |
| | $^{44}Ca$ | Laser ablation ICP-MS | 120 $\mu$m |
| **KCI** | Meltwater Conductivity, Insoluble Particles | Continuous Flow | > 0.7 |
| | Stable water isotopes ($\delta^{18}O$ or $\delta D$) | Discrete Sampling | 10 − 1.5 |

measured Ca signal have been shown to be in good correspondence with the lower resolution CFA Ca signal, as shown in Figure 3 and previously by Sneed et al. (2015).

### 3.2.1 Radiocarbon analysis

The measurements for radiocarbon dating of the ice core have been conducted at the Institute of Environmental Physics (Hei-
5 delberg, Germany) under close collaboration with the accelerator mass spectrometer (AMS) facility at the Klaus-Tschira-Lab in Mannheim, Germany. The microscopic particulate organic carbon fraction (POC) incorporated into the ice matrix was ex-tracted, combusted and analyzed for $^{14}C$ content. Calibration of the retrieved $^{14}C$ ages was performed using OxCal version 2.4 (Ramsey, 2016) and by convention the 1-sigma error range is shown (Stuiver and Polach, 1977). For details on the sample preparation and measurement procedure see Hoffmann (2016). The average ice sample masses were for both cores in a range
of ca. 300–500 g ice resulting in absolute POC masses below 10 $\mu$gC. For the KCC core, a fraction of the ice core with a cross section of 17 cm$^2$ was reserved for the POC $^{14}C$ analysis. Within the upper 44 m WE, a total of six samples were analysed, typically comprising between 40–60 cm of core. For the KCI ice core more core material (one third) was available, resulting in depth intervals of 40 cm length used for radiocarbon dating. Within the upper 40 m WE of the KCI core five samples have been analyzed so far.

## 4 Ice core dating

Ice core chronologies were established by annual layer counting as the main dating tool in combination with additional age constraints from $^{14}C$ for the lower core parts. For roughly the last 100 years, dated time horizons (1963 bomb-radioactivity, and the Saharan dust layers of 1977, 1947 and 1901, cf. Figure 5 below) are available to constrain the counting. The 1963 horizon was used to cross-check that the annual signal had been identified correctly (cf. sub-annual and multi-year signals). The dust
events were independently used for verification and typically lie within one to two years of the counted age scale (four years at maximum for the 1901 horizon). Regarding additional absolute age markers beyond 1901, the identification of volcanic





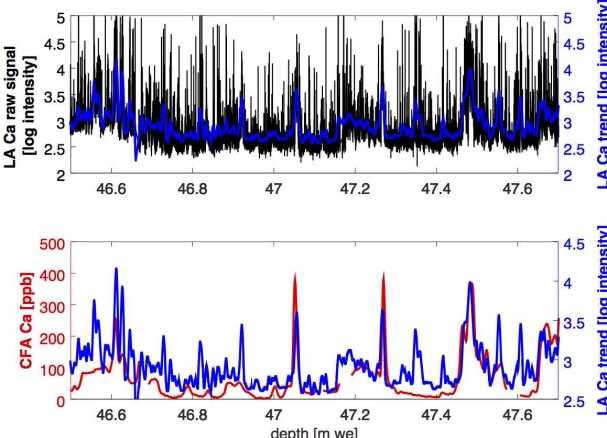

**Figure 3.** Ca signals obtained from the KCC ice core at around 65% relative depth using LA-ICP-MS and CFA in direct comparison after careful alignment of the two depth scales. Top row: Raw (black) and filtered LA-ICP-MS Ca signal (blue). Bottom: CFA Ca (red) vs filtered LA-ICP-MS Ca signal (blue). Note i) additional peaks and high-frequency information revealed by LA-ICP-MS, ii) a general agreement of CFA and low frequency LA-ICP-MS components is consistently observed over core parts measured by LA-ICP-MS.

eruptions solely based on basic ice chemistry profiles is not feasible at CG. This is due to the fact that the relatively weak signals of volcanic sulphate or volcanic acidity are easily overlooked at CG since they are embedded into the relatively large variability of Saharan dust associated sulphate (mainly from gypsum) and (acidity consuming) carbonate. More promising in this respect is the investigation of relatively volatile trace elements (Kellerhals et al., 2010b), or the detection of tephra markers, which are beyond the scope of this work, however.

### 4.1 The KCI chronology

For KCI, insoluble particle concentration and meltwater conductivity were used for annual layer counting, extending down to about 26 m WE. Below 26 m WE the identification of annual layers became ambiguous and was abandoned. This depth corresponds (taking the uncertainty in layer counting into account) to $(513 \pm 30)$ years before present (BP, referring to the drilling year of the respective ice core if not otherwise noted). A two-parameter model (based on a simple analytical expression for the decrease of the annual layer thickness with depth) was used to extrapolate a continuous age-depth relation to greater depth (Nye, 1963; Jenk et al., 2009).

### 4.2 The KCC chronology

All impurity species measured by CFA (Table 2) were used in combination for annual layer counting. Annual layers were defined as local maxima in at least two of the six impurity signals, with special emphasis on $NH_4^+$ featuring the largest seasonal amplitude. An example of counting annual layers in the CFA profiles in shown in Figure 4 a). In order to identify highly



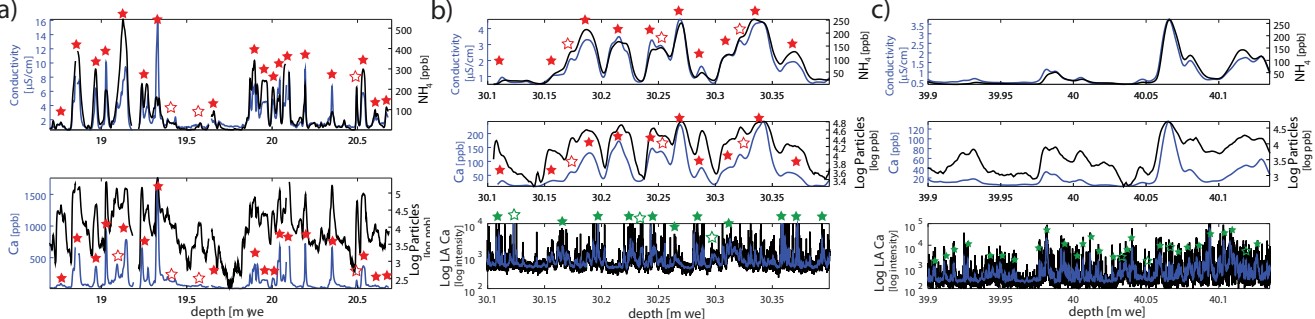

**Figure 4.** Examples for annual layer counting in KCC impurity profiles for three different depth sections, labeled a), b) and c), and corresponding roughly to 100, 250 and 1000 years BP, respectively (cf. Table 3). In the upper core parts (firn sections) CFA measured impurities were used for counting, with special emphasis on $NH_4^+$ (a). Counted years are marked as full stars, uncertain years as white stars. The middle row (b) shows an example of overlap in counting between CFA and LA-ICP-MS Ca, showing $(10 \pm 3)$ and $(11 \pm 3)$ years, respectively. The LA-ICP-MS Ca raw signal is shown in black together with Gaussian smoothing (blue). Note that a minor depth offset (at most a few cm) may exist between the CFA and LA-ICP-MS datasets. Counting within one of the deepest sections analysed for this study is shown in (c). Here, only the LA-ICP-MS Ca allows a reliable identification of almost sub-cm thin annual layers.

thinned, sub-cm annual layers expected to dominate the deeper core sections, an independent counting was established using the LA-ICP-MS Ca profile starting at 29.5 m WE (corresponding to 1760 AD). At this depth, the average annual layer thickness was estimated from CFA-based counting as around 3 cm. The LA-ICP-MS Ca record was investigated at full resolution and as a smoothed version (using the leading components in singular spectrum analysis or Gaussian smoothing). In its upper section,

the LA-ICP-MS Ca profile is characterized by regular occurrence of several distinct peaks grouped together going along with an elevated baseline of Ca concentration (Figure 4 b)). The groups of peaks are separated by a comparatively stable signal of low Ca concentrations. The latter is interpreted as resulting from the varying degree of winter snow being included in the record otherwise dominated by summer snow. Accordingly, the grouped peaks correspond to sub-annual snow deposition events of elevated Ca concentration during the summer period. For the depth interval 29.5–32.5 m WE, counting separated groups of

peaks (typically 3–5 peaks per annual layer) in the LA-ICP-MS Ca record results in good agreement with counting performed on the CFA profile (typically within ±1 year per 10 counted years, see Figure 4 b)). Below 32.5 m WE, average annual layer thickness becomes close to 1 cm and counting in the CFA profile becomes increasingly difficult (i.e. frequent "shoulder type" annual layers merged into a single impurity peak). The LA-ICP-MS Ca profile continues to show distinct groups of peaks that become increasingly closely spaced and eventually merge into single broad peak events (Figure 4 c)). Accordingly, LA-ICPMS

Ca was the dominant source of annual layer counting after around 32.5 m WE (1600 AD).

## 4.3 Age constraints from radiocarbon analysis

For KCC, results from $^{14}C$ analysis are found to back the annual layer counted age scale. Five of six $^{14}C$ dates agree with the counting within their 1-sigma range (Figure 5), corresponding to a root mean square deviation of 118 years (227 years





**Table 3.** Ice core age, dating uncertainty and annual layer thickness for selected depths

| Core | Depth [m WE] | Age [yr BP] | Uncertainty [yr BP] | Annual Layer Thickness [cm WE] |
|------|--------------|-------------|---------------------|--------------------------------|
| **KCC** | 10 | 42 | 1 | 18 |
| | 20 | 101 | 4 | 11.5 |
| | 30 | 251 | 12 | 2.7 |
| | 40 | 1013 | 72 | 1 |
| **KCI** | 10 | 88 | 4 | 7.4 |
| | 20 | 305 | 20 | 3.2 |
| | 30 | 693 | 62 | 1.7 |
| | 35 | 1066 | 77 | 1 |

including the outlier). The outlier [14]C point contradicts a monotonic increase of age with depth and is thus disregarded. This is justified, because the [14]C age of this sample matches with a very sensitive section of the [14]C-calibration curve. Therefore already a small, unknown blank contribution would be able to shift the calibrated age of this sample significantly. Within the 2 sigma error range it also hits the error range of the annual layer counting chronology in the present configuration. Its deviation is therefore not of consequence.

For the KCI ice core, the radiocarbon ages are found to agree with the extension of the existing age scale based on the two-parameter model. It seems worth noting, however, that four out of five [14]C points lie systematically above the extrapolated age scale (albeit in agreement within their 1-sigma range). Only the sample at 28.4 m WE shows an age that is significantly older than expected. This can on one hand be due to the extremely small (also compared to the other KCI samples) sample size of only 2.2 $\mu$gC making this sample prone to even very small potential blank contributions. In this context also a potential influence of aged organic material (e.g. from Saharan dust) has to be regarded. At present, the age of this sample is therefore regarded as an outlier. Additional radiocarbon measurements of this core section above and below the critical sample are planned to further refine the match, and to test if the systematic deviation of the [14]C ages persists.

### 4.4 Dating uncertainty

Potential sources of uncertainty in annual layer counting stem from i) erroneously identifying or missing of existing annual layers, ii) interpolating data gaps and iii) an incomplete stratigraphy missing years due to annual snowfall fully eroded from the surface. Regarding i), we estimated the likelihood of miscounting layers by marking "uncertain years" (Figure 4). In view of the high snow erosion at CG, uncertain layers were defined as additional peaks in close proximity to an annual layer (e.g. "shoulder type" peaks). To quantify counting uncertainty from uncertain layers, we followed the approach successfully employed for Greenland ice cores. This is to count uncertain layers as 0.5±0.5 years and to estimate the maximum counting error (MCE) from N uncertain layers as N x 0.5 years (Andersen et al., 2006; Rasmussen et al., 2006). With 144 uncertain layers detected within the upper 40 m WE of KCC, this corresponds to an uncertainty of ±72 years at 1013 years BP . With





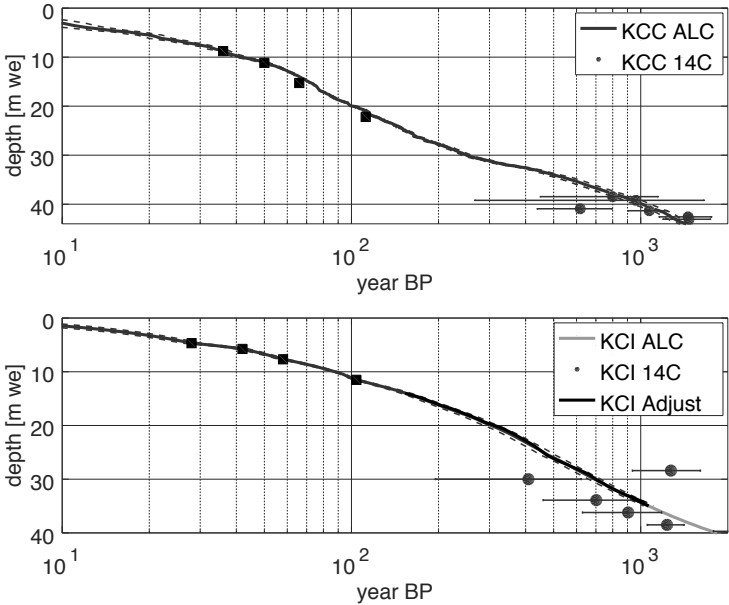

**Figure 5.** Age-depth relations over the last 1000 years for KCC and KCI, shown in the top and bottom row respectively. Age is plotted on a logarithmic axis, together with the according estimates of maximum dating uncertainty (dashed lines) and $^{14}$C age constraints (with 1-sigma range) for KCC and KCI. Also shown is the adjusted age scale of KCI based on the stable water isotope time series comparison (solid black line, within less than 15 years of the original dating and thus hardly distinguishable here, see text). Absolute dating horizons used roughly within the last 100 years (see text) are shown as black squares. Note that the KCI chronology is based on a simple extrapolation below 26 m WE and has large uncertainty beyond 1000 years BP (thus indicated as light gray line only).

respect to ii), the depth interval considered in this work was completely recovered without core loss. The ends of the CFA core sections were trimmed in case of irregular core breaks. This resulted typically in less than a centimeter of missing CFA data, thus not interfering with annual layer counting. The ECM profile was used as an alternative backup across these short CFA data gaps. Likewise, the CFA data was used as an alternative indicator where the LA-ICP-MS profile was incomplete, which

5  only concerned one major instance of missing LA-ICP-MS data between about 33.8–34.24 m WE.

Contribution iii) constitutes a fundamental difference relative to Greenland conditions, since CG is not a closed system with respect to precipitation and loss of the annual snowfall in selected years can occur. The frequency of occurrence in these total snow loss events is, however, extremely hard to quantify. Counting annual layers in between the above mentioned (dust) horizons within the last century, reveals an offset of typically only one to two years as compared to the known age of the

10  horizons. Thus, the counting appears not to be systematically flawed by missing years. Hence we regard uncertainty i) as dominant and use the MCE as a uncertainty estimation of the KCC age scale.

The uncertainty of the KCI age scale was obtained in a consistent manner, using the MCE for the annual layer counted interval





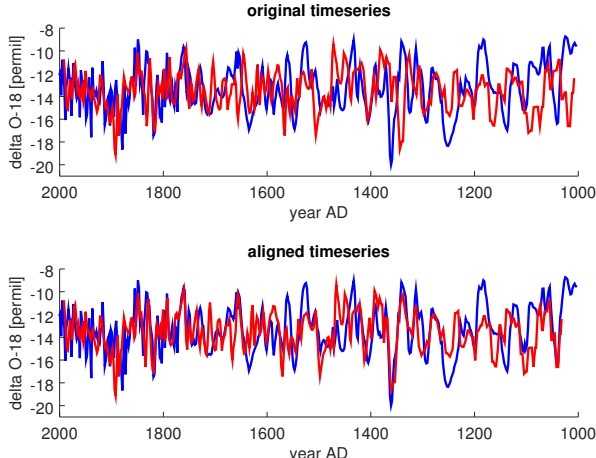

**Figure 6.** Stable water isotope time series of KCC and KCI, shown in blue and red, respectively. The top row shows both records on their original time scale over the last 1000 years. The KCI age scale was adjusted using the algorithm of (Lisiecki and Lisiecki, 2002) to optimise the match with KCC.

and extrapolating the upper and lower uncertainty limits with the two-parametric model. Figure 5 shows the resulting age-depth relation and uncertainty bands for KCC and KCI, together with $^{14}$C dates (shown with their 1–sigma uncertainty range) available for the respective depth interval. Table 3 gives a complementary summary of the age-depth relation, uncertainties and annual layer thickness for 10 m depth intervals. It is important to note that we are less confident about the age-depth relation

of KCI compared to KCC, due to KCI featuring i) annual layering counting using only two bulk parameters and only down to 26 m WE and ii) the extrapolation by the two-parameter model.

### 4.5 Inter-core time series comparison and age scale alignment

To investigate potential offsets between the KCI chronology and the presumably more reliably dated KCC, we compared the stable water isotope time series of the two cores. This is motivated by the fact that the decadal isotope trends among the

10 CG ice cores have been previously shown to agree over the last 250 years (Bohleber et al., 2013). Without substantial dating offset between the cores, this inter-core agreement should hold also on longer time intervals. It is important to note that due to the strong effect of isotope diffusion at CG, inter-annual or even seasonal isotope variability is effectively eliminated. As a consequence, the records (except for the last 100 years in KCC) resolve only decadal-scale variability at best. Hence we did not apply any further smoothing to the time series. In order to avoid potential biases from increasing sampling resolution, both

time series were sub-sampled to nominal biennial resolution. Figure 6 shows the comparison of the respective time series on their original time scales for the last 1000 years BP.

The two original time series of KCC and KCI already feature striking similarities, although frequently separated by a lag between the two time series (e.g. note the distinct isotope minima around 1360 AD). The direction and magnitude of this lag



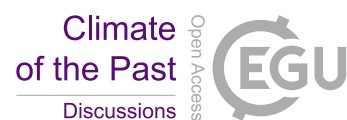

varies with time, hampering an absolutely straightforward adjustment to match the two records. Aiming to adjust the KCI record to KCC time series, we employed the powerful algorithm developed by Lisiecki and Lisiecki (2002) for correlating paleoclimate time series. In doing so, we left the last 150 years of the KCI age scale unchanged (since considered reliably dated) but did not prescribe any further user-defined tie-points to the algorithm. The result shows the original lag between the

two time series eliminated (Figure 6). A maximum shift of around 15 years is needed to align the two records (e.g. around the 1360 AD isotope minimum), which is within the estimated dating uncertainty of KCI (Figure 5). As a result, the aligned time series are significantly correlated ($r = 0.47$). This degree of correlation is within the typical range of correlating CG isotope time series on the decadal scale within the last 250 years (Bohleber et al., 2013). However, this is the first time that the correlation holds to this extent also for the comparatively old core sections, e.g. we find a correlation coefficient of $r = 0.50$

when considering the interval 1500–1000 AD only. In the following, all KCI time series are considered on their aligned time scale.

## 5   Results and Discussion

The age scale of KCC provides the first chronology of the last millennium for a CG ice core that is fully based on annual layer counting. The new KCC age scale offers the to-date most accurate foundation to study the CG proxy time series over long

time scales, e.g. regarding the recent investigation pursuing the link with historical evidence by More et al. (2017) who used a slightly adjusted version of the age scale presented here (albeit not significantly different with respect to uncertainty). The novel technique of LA-ICP-MS was crucial for a reliable identification of cm and sub-cm thin layers in the deeper parts of the core. Thereby, this work adds to recent studies (e.g. Della Lunga et al., 2017; Haines et al., 2016; Mayewski et al., 2014) to demonstrate the potential of the high-resolution impurity records afforded by LA-ICP-MS for investigating highly thinned

sections of polar and alpine ice cores. The combination of high-resolution annual layer counting and radiocarbon analysis promises a break-through also for dating highly thinned deep parts of ice cores drilled at other sites.

### 5.1   Stable water isotope records

The covariation of the $\delta^{18}$O time series between KCI and KCC strongly suggests a common atmospheric driver, i.e. temperature. At first glance Figure 6 shows an increasing trend over the last 100 years but also generally higher mean isotope levels

prior to about 1900 AD. A quantitative use of the common isotope signal would therefore require addressing systematic so-called "upstream effects" and calibration of the isotope signal against instrumental temperature.

Upstream-effects concern the systematic variation in seasonality of the net accumulation upstream of the drilling site and have the potential to bias long-term core averages. Quantifying this effect requires accurate identification of the upstream catchment area (typically by sophisticated flow modeling) and evaluating the spatial variability in mean isotope levels. Dedicated efforts to

evaluate the upstream-effect for the KCI-KCC flow line are currently underway (pers. comm. Carlo Licciulli IUP Heidelberg). From a preliminary inspection of snow pit data recently obtained for the KCI-KCC flow line, there is no clear indication of a systematic trend in mean $\delta^{18}$O levels upstream of KCC, however.



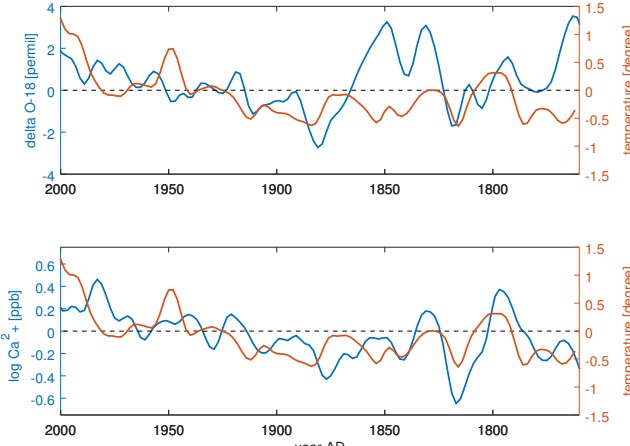

**Figure 7.** Comparison of KCC $\delta^{18}$O and Ca$^{2+}$ against the CG modified instrumental temperature (orange), shown in top and bottom, respectively, and covering the full instrumental period back to 1760 AD. Shown are anomalies relative to the respective 2000-1860 AD mean, and as decadal trends obtained from Gaussian smoothing.

In order to calibrate the stable water isotope signal, we used the instrumental temperature dataset compiled in an earlier study (Bohleber et al., 2013). This temperature dataset (referred to here as "CG modified temperature") was specifically adjusted to the CG ice core conditions, taking into account the summer bias in precipitation and snow deposition. To calculate an isotope/temperature sensitivity, we considered both KCC and KCI individually as well as a stack of the two stable isotope records

(calculated as their simple average). Using 2000–1860 AD as calibration period (thus deliberately avoiding the "early instrumental period" prior to 1860) our results reproduce earlier findings of Bohleber et al. (2013). This specifically includes showing i) an overall agreement between the isotope and temperature record interrupted by characteristic decadal mismatch periods (Figure 7), ii) an increase in isotope/temperature correlation for multi-annual and decadal averages (e.g. $r = 0.33, 0.47, 0.64$ for discretely binned annual, 5 and 10 year averages, respectively in case of KCC) and iii) higher correlations obtained from

the stack vs. the individual time series (e.g. $r = 0.48, 0.67, 0.79$ for annual, 5 and 10 year averages, respectively).

Regarding sensitivity values, we also find an increase with length in averaging period as well as substantially higher sensitivity values for KCI than KCC, revealing 2.3 vs. 1.4 ‰/°C, respectively, when using discretely binned 10-year averages (and 1.8 ‰/°C for the stacked record). Hence we obtain sensitivity values about threefold of what is expected, e.g. based on the isotope/temperature relationship of 0.65 ‰/°C reported by Rozanski et al. (1992) for European temporal trends in precip-

itation. In addition to the choice of time scale (length of discrete averages), changes in snow preservation are expected to bias sensitivity (cf. the conceptual consideration in section 2.2). This is consistent with the sensitivity difference among KCI and KCC, since an even more strict confinement towards sampling the high summer season can be expected for the lower accumulation KCI. It is important to note that the high isotope-sensitivity deserves a separate thorough investigation, ideally comprising regional climate-isotope modeling, which is outside the scope of this study. Until an adequate long-term calibration



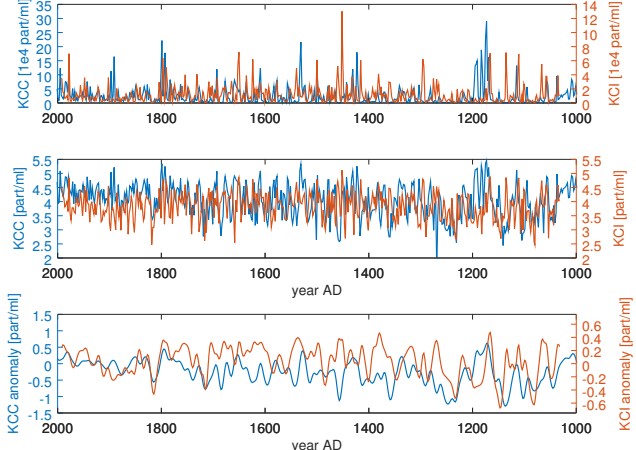

**Figure 8.** Inter-core comparison of the insoluble particle signal of KCC (blue) and KCI (orange). The top and middle row show the insoluble particle time series on a linear and logarithmic scale, respectively. The bottom row shows decadal trends of anomalies with respect to their 2000–1860 AD mean, highlighted by Gaussian smoothing. The KCI record is on the adjusted time scale after matching the stable water isotope records.

of the CG isotope signal is achieved, however, we do not attempt a quantitative temperature reconstruction based on the isotope composite record so far.

## 5.2 Mineral dust proxy records

Two kinds of signals are investigated here from the mineral dust related proxy time series: i) the long-term average $Ca^{2+}$
concentration, and ii) the frequency of occurrence in Saharan dust deposition. The latter is expected to be comparatively more robust against snow preservation influence.

In order to investigate to what extent the time series agreement observed for $\delta^{18}O$ also holds in case of mineral dust related species, we use the insoluble particle signal as a surrogate for $Ca^{2+}$ (since $Ca^{2+}$ has not been measured for KCI). Figure 8 shows an overview of the insoluble particle datasets of KCC and KCI. KCC data shows that the insoluble particle signal
and $Ca^{2+}$ concentrations are generally highly correlated (e.g. $r = 0.9$ within the last 1000 years). The KCC-KCI inter-core comparison of insoluble particle records reveals agreement of decadal scale features, as well as similarities regarding periods of low concentrations and higher peak abundance (e.g. 1800–1820 vs. 1780–1800, respectively). Differences in the magnitude of individual peak events as well as mean levels of particle concentrations can be explained in light of i) the KCI particle signal measured on diluted sample meltwater, ii) potential calibration differences in the optical particle sensor and iii) inter-site snow
deposition variability. Accordingly, it was not attempted to construct a composite record of insoluble particles from the two cores.





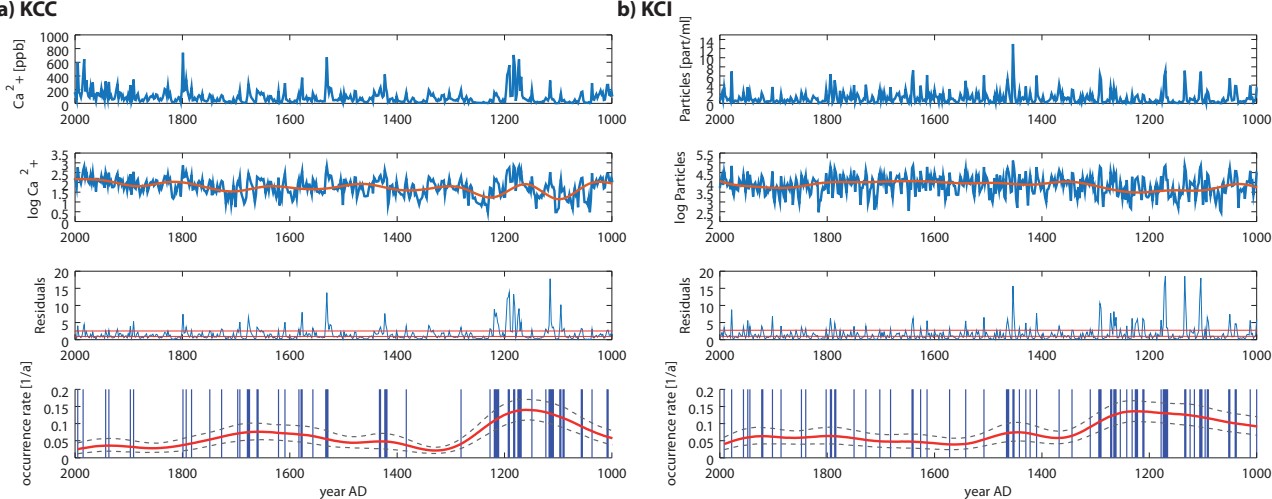

**Figure 9.** Results from detecting Saharan dust events in the ice core records and estimating their long-term frequency of occurrence. Figure part a) corresponds to KCC, part b) to KCI (see text). The bottom row shows the detected events (blue) and the frequency of occurrence kernel estimate with a 51-year bandwidth (bottom rows, in red) together with 90% confidence intervals.

### 5.2.1 Detection of Saharan dust peak events and frequency of occurrence

Essential for the calculation of a robust occurrence rate of dust events are adequate means to distinguish desert dust from background and from deposition events of long-range transported anthropogenic pollutants. While the particle signal alone is not sufficient for differentiating these events, Saharan dust layers in CG ice cores can be reliably identified based on the

analyses of $Ca^{2+}$, supplemented by alkalinity measurements and, in principle, particle size distribution (Wagenbach et al., 1996). The central criteria used in this study in order to identify Saharan dust are strongly elevated concentrations of $Ca^{2+}$ coinciding with acidity values reduced to alkaline levels. Since no direct acidity measurements are available in our case from CFA, we rely on the ECM record for this purpose. At CG high dust levels are able to reduce the ECM signal to almost zero (rendering the ECM to be a qualitative dust indicator rather than an quantitative acidity gauge). Dust anomalies were identified

as "peaks over threshold". A robust spline smoothing was used to remove the general trend from the $Ca^{2+}$-data. Peak events then needed to exceed three times the median absolute deviations (Figure 9). We have also used the particle size distribution to investigate exemplarily a small number of dust events, finding that dust events show systematically higher CPP with respect to dust-free core sections.

To detect the frequency of occurrence in dust peak events, we followed the statistical tool outlined in Chapter 6 of Mudelsee

(2010). For a non-parametric occurrence rate estimate we used a moving Gaussian kernel (bandwidth 51 years) and accounted for boundary effects. For KCC, only the subset of peak events coinciding with a vanishing ECM signal was considered to be of Saharan dust origin. For KCI, we employed the same peak detection scheme to the insoluble particle signal. However, due to the lack of a full ECM profile, no subset corresponding to low acidity could be defined. Using a direct comparison with





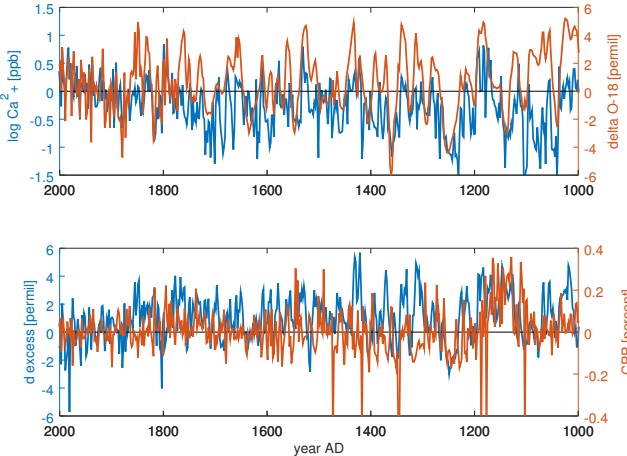

**Figure 10.** Records of KCC, displaying $\delta^{18}$O together with Ca$^{2+}$ (top row) and deuterium excess together with the coarse particle percentage (CPP, bottom row), shown in blue and orange, respectively. All time series are shown as anomalies relative to the respective 2000–1860 AD mean. Note the high levels of deuterium excess and CPP around 1100–1200 AD.

the insoluble particle signal of KCC, with and without ECM correction, we found that the respective uncorrected frequency of occurrence is expected to contain a minor bias towards higher peak abundances, but leaves the overall features unchanged. As the main robust features for KCC and KCI, the frequency of occurrence in dust peaks is systematically increased prior to 1250 AD with respect to the rest of the record (Figure 9). Dust anomalies are found clustered in periods around 1100–1200 (extending into the 1200s), around 1400–1450 and, for KCC only, between 1500–1800. This is in broad agreement with periods of enhanced Saharan dust deposition reported by Thevenon et al. (2009) obtained from elemental analysis in a CG ice core, albeit at much coarser resolution and larger dating uncertainty. Notably we also recognize a large dust peak located between 1780 and 1800 suggested as a dating reference horizon by Thevenon et al. (2009).

## 5.3 The CG isotope and mineral dust proxies combined

We now combine isotope and dust proxies and add evidence from dust frequency analysis and secondary quantities of deuterium excess and coarse particle percentage. Figure 10 shows the time series of Ca$^{2+}$, $\delta^{18}$O, deuterium excess and CPP. Due to the logarithmic distribution of the Ca$^{2+}$ data, we generally use a log-scale to show the Ca$^{2+}$ time series. For KCC, we find $\delta^{18}$O and Ca$^{2+}$ to be significantly correlated over the last 1000 years (at $r = 0.49$, cf. Figure 10). This suggests that the common decadal-scale signal driver behind the shared variability between the $\delta^{18}$O time series of KCC and KCI also holds for Ca$^{2+}$. Prior to about 1860 AD, the $\delta^{18}$O time series constitutes nearly an upper envelope signal as compared to Ca$^{2+}$. Short excursions to low Ca$^{2+}$ concentrations missing a respective counterpart in $\delta^{18}$O may have been smoothed out by isotope diffusion. Worth noting in this respect, the firn-ice transition in KCC coincides roughly with the last 100 years in the record (Tables 1 and 3).



### 5.3.1   The long-term temperature significance of the Ca$^{2+}$ records

Within the calibration period 2000–1860 AD, we find the overall increasing trend in instrumental temperature to be represented also in increasing levels of $\delta^{18}$O and Ca$^{2+}$ (Figure 7). The Ca$^{2+}$ signal correlates significantly with the CG modified instrumental temperature at $r = 0.41, 0.56, 0.71$ using biennial, 5 and 10 year averages, respectively, within the calibration period.

Nearly identical correlation values are obtained for the full instrumental period back to 1760 AD. Within the calibration period (Figure 7), we compared the decadal trends (highlighted by Gaussian smoothing) of the CG modified instrumental temperature with $\delta^{18}$O and Ca$^{2+}$, respectively. The comparison reveals that the Ca$^{2+}$ signal performs similarly to $\delta^{18}$O in explaining variance of the temperature data (both at around 25%, although only interpreted with caution due to the autocorrelation of the smoothed curves).

On this basis, the potential of the Ca$^{2+}$ signal to qualitatively record temperature variability is explored further. The biennial logarithmic Ca$^{2+}$ is calibrated tentatively against instrumental temperature using linear regression within the time period 2006–1860 AD. The respective 90% confidence intervals are used to calculate a temperature reconstruction with uncertainty bands (0.7–1.8 $^\circ C$/log Ca$^{2+}$ [ppb]). Decadal trends are again highlighted by Gaussian smoothing in Figure 11. The resulting 1000-year record is shown with a tentative 200-year extension. Regarding its overall features and in view of remaining

dating uncertainties, the record provides evidence of "Little Ice Age conditions" systematically cooler than the reference and calibration period, with an average of $-0.3^\circ$C between 1800–1200 AD. A shorter warm interval of about $+0.3^\circ$C is found in the late 1100s.

### 5.3.2   Medieval outstanding period

Being the most outstanding period in the dust event occurrence warrants taking a closer look at 1100–1200 AD. This outstand-

ing period is characterized by i) an increased frequency of Saharan dust events, ii) its above average levels, both in CPP and deuterium excess, starting to rise around 1100 AD, and iii) a delayed relative increase in $\delta^{18}$O and Ca$^{2+}$, e.g. first showing minimum concentrations between 1130–1170 AD followed by a shorter maxima around 1170–1200 AD. The CPP maximum constitutes the dominant feature of the entire record. The median of the CPP record within this 1100–1200 AD time period ($0.61 \pm 0.11$, reported with one median absolute deviation) indicates an increase in coarse particles by about 12% relative to the

median of the rest of the 1000-year time period ($0.48 \pm 0.05$). The connection of the distinct increase in coarse particles with enhanced dust event frequency indicates an increase in the direct transport of Saharan dust (as opposed to indirect advection with longer pathway and thus stronger decrease in coarse particles). Notably, this view is consistent with increased deuterium excess, which would be expected from warm and dry air masses collecting moisture over the Mediterranean.

### 5.4   Comparison with other proxy reconstructions

We further explored our Ca$^{2+}$-based temperature-reconstruction attempt in comparison with other proxy reconstructions of European summer temperature. We show here results from using the mean European summer temperature anomalies reconstructed by Luterbacher et al. (2016), considering their composite-plus-scaling method (CPS) adjusted to biennial resolution



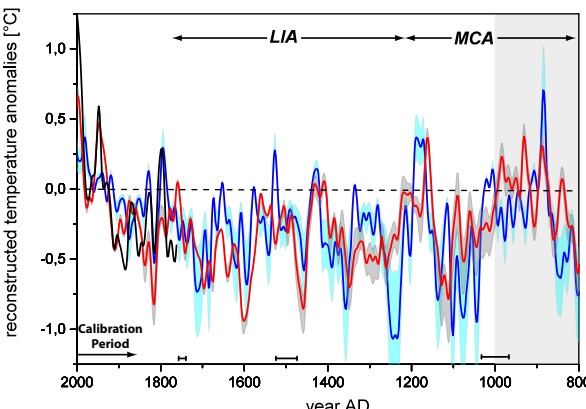

**Figure 11.** Comparison of decadal temperature trends as anomalies with respect to the mean of 2006–1860 AD. Shown are calibrated temperatures obtained from the KCC $Ca^{2+}$ variability (blue lines, with uncertainty indicated as light blue bands). Also shown are instrumental temperature data (black) and the summer temperature reconstruction of Luterbacher et al. (2016) in red (uncertainty as gray bands). Note that the overall co-variation among the two reconstructions persists for at least another 200 years beyond 1000 AD (light gray shaded area). Black bars on the bottom indicate maximum dating uncertainty.

and our reference time period of 2006–1860 AD. The decadal trends (represented by Gaussian smoothing) of both reconstructions shown in Figure 11 are generally consistent in their overall features (and formally correlate at $r = 0.4$). These features comprise the recent warming trend and below average conditions during the "Little Ice Age", and a warm episode in the late 1100s. The overall agreement is especially noteworthy in the light of the remaining dating uncertainty of KCC, which may be

the cause of smaller offsets between the two records. The only major feature of disagreement occurs around the already noted minimum around 1250–1230 AD. The overall low levels of impurities (especially $NH_4^+$) and $\delta^{18}O$ may point to increased deposition of winter snow during this time, or exceptionally cold summer conditions. It is worth noting that if, tentatively, extending the comparison for another 200 years beyond 1000 AD, the agreement between the two reconstructions continues to last until 800 AD, consistently showing a relatively warm interval lasting between about 1000–850 AD.

Regarding the medieval period 1200–1100 AD, which stands out in the mineral dust proxies, the picture of increased meridional transport favoring direct Saharan dust advection over the Mediterranean is consistent with a NAO+ dominated MCA, lasting 1100 to about 1300 AD, as proposed by Trouet et al. (2009). Although NAO is mainly a winter signal (thus not a direct concern to CG summer representative ice core signals), positive NAO phases (stronger Azores High) comprise stronger westerlies and wetter winters over northwestern Europe and decreased precipitation over southern Europe and northwestern

Africa. An NAO+ phase thus entails i) a relative increase in meridional transport over the Mediterranean in favor of increased dust advection and ii) dry conditions over the Mediterranean and Northern Africa, in favor of increased dust mobilization.





## 6 Conclusions and Outlook

A combination of state-of-the-art methods in ice core analysis allowed us to date the latest CG ice core KCC with unprecedented confidence. The breakthrough in this respect was to extend annual layer counting, for the first time at CG, over more than 1000 years BP and finding the resulting age scale corroborated by radiocarbon analyses. To further extend our investigation, we will apply our new dating approach also for the bottom 10 m WE of KCC. The combination of high-resolution annual layer counting afforded by LA-ICP-MS with constraints from radiocarbon analyses could be employed with great success also at deep sections of other (mountain) ice cores. By means of the improved age scale it became possible, for the first time, to demonstrate that the inter-core agreement in decadal isotope variability among two cores on the same flow line extends over the last 1000 years. The inter-core agreement indicates a common driver of the shared signal, also extending to the long-term variability in $Ca^{2+}$. We find that a coupling between a direct atmospheric temperature signal and systematic snow preservation changes is likely the explanation for the observed co-variation between $\delta^{18}O$, $Ca^{2+}$, and instrumental temperature. The intrinsic contribution of snow preservation may bias the isotope-temperature sensitivity and at present hampers the quantitative use of the isotope-thermometer at CG, thus motivating further future investigation of this enigmatic effect. Exploiting the $Ca^{2+}$-temperature agreement offers an alternative for a quantitative reconstruction, and i) proves to be consistent with other latest summer temperature reconstructions, and ii) reproduces overall features regarding the "Little Ice Age" and the "Medieval Climate Anomaly". Parameters less influenced by snow preservation (dust event occurrence rate and particle size distribution) reveal an exceptional medieval period around 1100–1200 AD, indicating a relative increase in meridional flow and dry conditions over the Mediterranean, which is consistent with the existing view of a NAO+ dominated medieval climate period. These details on atmospheric circulation add to our investigation of long-term temperature variability, and ultimately fully demonstrate the potential of the CG ice cores for contributing to future efforts at multi-proxy climate reconstructions.

*Acknowledgements.* We are grateful to numerous colleagues for their commitment regarding field work, ice core drilling and ice core analyses. In particular we would like to acknowledge the support of the Initiative for the Science of the Human Past at Harvard University and all its project members. Additional invaluable support in ice core processing was provided by the Alfred-Wegener-Institute, Helmholtz Center for Polar and Marine Research, Bremerhaven. The Klaus-Tschira-Lab Mannheim is acknowledged for their support in radiocarbon analysis. We also would like to thank Johanna Kerch, Carlo Licciulli, Josef Lier and Lars Zipf from IUP Heidelberg for their support. Recovery and analysis of the 2013 CG ice core KCC were supported by the Arcadia Fund of London (AC3450) and the Helmholtz Climate Initiative REKLIM. Work on the 2005 CG ice core KCI has been funded by the European Union under contract ENV4-CT97-0639 (project ALPCLIM) and within the project ALP-IMP through grant EVK2-CT2002-00148. LA-ICP-MS ice core analyses were conducted in the Climate Change Institute's W. M. Keck Laser Ice Facility at the University of Maine supported from the W. M. Keck Foundation and the National Science Foundation (PLR-1042883, PLR-1203640). Financial support was provided to P.B. by the Deutsche Forschungsgemeinschaft (BO 4246/1-1). We would like to especially thank and acknowledge our late colleague Dietmar Wagenbach (Heidelberg University) for his long-standing contributions to glaciological research at Colle Gnifetti, and in particular for sharing his unique expertise with us at the early stage of our project.





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
