# Peer review of "Temperature and mineral dust variability recorded in two low accumulation Alpine ice cores over the last millennium"

_Climate of the Past, 2017_

## Referee Comment (RC1) · Anonymous Referee #1 · 17 Jul 2017

This paper presents an exceptional data set composed by stable isotopes and various proxies of aeolian dust deposition as measured in two Alpine cores from Colle Gnifetti (European Alps) during the last 1000 years. The quality of the data presented appears robust and, in general, the time scale adopted seems reliable. However, while this data set does deserve publication, the interpretation and use of the results obtained is often weak, speculative and potentially misleading. I believe the discussion section of the journal Climate of the Past represents a suitable venue to possibly sort out several important interpretative issues before the possible final publication.

Regardless the source of the dust deposited, the correlation between $Ca_{2+}$ (as dust

proxy) and atmospheric temperature may be really linked to a poorly known post depositional process. Dust in surface snow may indeed facilitate a metamorphic process that could consolidate and conserve the snow on site, for instance via a reduced albedo (larger amounts of solar radiation absorbed) at times with more frequent clear sky and higher atmospheric temperatures. In my view the authors should really try to better describe this possible process.

Considering the poor seasonality of intense dust events of Saharan origin, at this time I do not see how their semi-quantitative model can be useful to support the possible occurrence of out of phase dust (wet) deposition and post-depositional consolidation. Most importantly, the out of phase dust/snow deposition and the possible post-depositional process linked to atmospheric temperature is not sufficiently supported by data. In this respect results from past snow pit studies might be illuminating and should be extensively presented.

While the occurrence of a Saharan dust fallout via wet deposition is very well established at Colle Gnifetti, the interannual variability of this kind of Saharan events is very high in the Alps and is mostly related to periods when atmospheric advection dominates (late fall, winter, early spring). During summer, however, atmospheric vertical convection rules in the Alps and dry transport and deposition of dust of local/Alpine origin is extremely likely and probably intense also on Colle Gnifetti. Source reconstruction of dust entrapped in ice cores is very complex even when sophisticated multiple proxies are used (e.g Sr and Nd isotopes). Ca2+, even when combined with ECM and dust size, cannot be considered a specific proxy of Saharan dust events and cannot discriminate with sufficient confidence between different kinds of sources (e.g. Saharan vs. local) and atmospheric circulation (e.g. meridional flows vs. vertical convection).

Remarkably, if Ca2+ concentration is assumed to really trace dust from a specific area (e.g. Sahara desert) this would severely prevent its general use as paleothermometer as this parameter would strongly depend on environmental conditions at the source that are not directly related to atmospheric temperature. For instance a Ca2+ based paleo

thermometer might be severely biased by different soil conditions at the source, due for instance to changes in atmospheric circulation and precipitation, not temperature (e.g. green Sahara during the middle-early Holocene). The use of Ca2+ to construct a pale thermometer could thus depend on very site specific post-depositional processes and on changing environmental conditions at the source of dust over time. Thus it is fundamental to greatly caution about its general use in space (drilling site dependence) and time (it may work only during certain times). In conclusion, while the relation between Ca2+ and temperature is interesting, its extended use cannot provide unambiguous novel knowledge about past atmospheric temperature.

When compared to Ca2+, the use of stable isotopes as a proxy of atmospheric temperature is better justified by the physics and is less sensitive to source effects due to moisture changes. It is remarkable that the correlation of stable isotopes and instrumental atmospheric temperature is very good during the last 150 years when the recorded instrumental temperatures are most robust. It appears that the real problem of the possible paleothermometer based on stable isotpes is the "excessive" and spatially variable sensitivity (1.4-2.3 per mill/C) when compared to what one would expect (0.65 per mill/C). For this reason the authors decide not to attempt a quantitative temperature reconstruction based on stable isotopes. However, while this decision may be justified, I believe that an extended discussion of the possibly biased paleo-temperatures obtained by means of this presumably too sensitive/variable paleothermometer would be very interesting. This could show for instance the inconsistency of the temperatures obtained, even considering physical processes such as the amplification of atmospheric temperature anomalies with the elevation.

A likely reliability of the timescale obtained for the two cores from Colle Gnifetti is suggested by the good correlation of Ca2+ with an independent, well dated, past summer temperature record obtained from tree rings. However, the reliability of this time scale may be just due to the use of recent absolute time horizons (during the last century) and 14C measurements (although the significant reference, PhD thesis of Hoffmann 2016,

cannot provide an accessible and peer reviewed methodological support). Counting of annual layers is not convincing in the deep part of core KCC. In particular the concept of "group of peaks" of Ca2+ to identify a single annual layer seems extremely arbitrary, and, at this time, not supported in the paper by any snow pit observation. In addition a possibly larger deflation of lighter snow during the colder summers of the Little Ice Age may have removed more annual layers that expected. In this way, it is very possible that the interval period between 150 and 600 BP remains unconstrained and prone to larger uncertainties.

While Colle Gnifetti is a very well know ice core drilling site and many glaciological studies have been performed, this paper fails to offer the necessary comparison with existing data set available from other ice cores (and snow pit samples) obtained at the same site. In particular several studies of particulate and aerosol deposition were performed (e.g. Thevenon, JGR 2009) and should be carefully compared to check the consistency (or not) of the new findings with the previous results.

Specific comments:

P1 L1: "Among ice core drilling sites in the European Alps, the Colle Gnifetti (CG) glacier saddle is the only one to offer climate records back to at least 1000 years"

There is now in the Eastern Alps a new ice core climate record that goes back almost 7000 years (Gabrielli et al. The Cryosphere 10, 2779–2797, 2016; Gabrielli et al. 19th EGU General Assembly, EGU2017, proceedings from the conference held 23-28 April, 2017 in Vienna, Austria., p.9932).

P1 L8: "A high and potentially non-stationary isotope/temperature sensitivity limits the quantitative use of the stable isotope variability thus far".

This statement is not discussed sufficiently within the text.

P1 L15: "the medieval climate period around 1100–1200 AD stands out through an increased occurrence of dust events, potentially resulting from a relative increase in

meridional flow and dry conditions over the Mediterranean".

While the frequency of the dust horizons is reproducible in the two cores, thye cannot be linked to individual dust events of Saharan origin that cannot be unambiguously distinguished from the occurrence of local past summer surfaces marked by dust accumulation.

P2 L5: "Colle Gnifetti (CG) in spite of its limited glacier depth – stands out as the only site where net snow accumulation is low enough to provide records over the last millennium and potentially beyond at a reasonable time resolution".

Again, this is not correct, please see Gabrielli et al. The Cryosphere 10, 2779–2797, 2016.

P4 L5 "A single deposition event typically lasts less than a few days (Sodemann et al., 2005; Schwikowski et al., 1995). The associated warm air temperature and the substantially lowered snow albedo both support surface snow consolidation and partly protect the dust layer from wind erosion."

As long as air temperature is below the freezing level (as during snow events), this cannot be a factor facilitating snow consolidation.

P4-10 "Therefore, the $Ca^{2+}$ record of the CG ice cores is primarily related to mineral dust and dominated by Saharan dust related spikes".

This conclusion is unsupported by more recent data detailing more specific proxies of Saharan dust.

P4-27 "For instance, warm summers feature increased vertical mixing and hence a higher atmospheric impurity load, and in addition, entail faster fresh snow consolidation. This may lead to an increased relative amount of impurity-rich summer snow deposition."

This is a very reasonable and, unlike the meridional Saharan advection, a more regular

process in the Alps. I'm not sure why the authors do not consider and discuss it further within the text.

P5-L7 "Here we follow the model of Wagenbach et al. (2012), which assumes sinusoidal cycles for the precipitation-borne signal S(t) and the surface accumulation pattern A(t), and a phase-lag t$\varphi$ between S(t) and A(t)."

While accumulation and delta 18O seasonal patterns are accepted at Colle Gnifetti, it is much less so for a disturbed seasonal signal like dust (Ca2+). The authors need to present data supporting how the sinusoidal assumption is justified for Ca+2 deposition.

Paragraph 2.2 In general I do not find that this paragraph well written or even necessary. It is not clearly explained what are the main motivations and conclusions of the conceptual model. At this time I'm also not sure how it supports the rational of the interpretation. Does the model show that a phase lag between deposition and consolidation explain some recent observations? If so, the phenomenology of these processes needs to be supported, displaying existing data (e.g. snow pits) that could complement the conceptual discussion performed by means of this model.

P7-L2 "The threshold (4.0 $\mu$m. ) was chosen such that it corresponds to the expected median particle diameter of Saharan dust particles at CG (Wagenbach and Geis, 1989)."

Is this threshold necessary AND sufficient to discriminate between Saharan and non-Saharan dust? I do not think so. The reference reported is pretty old and in the meantime many tools have been developed to characterize dust sources. In my view this threshold is just indicative but not strictly discriminant.

3.2.1 Radiocarbon analysis. A table detailing all the results obtained by analyzing the 6+5 samples from KCC and KCI needs to be reported, including the linked uncertainties.

4 Ice core dating. A Table summarizing all the time horizons used in KCC and KCI

needs to be reported. Another table indicating different kind of annual layer counting in different sections of the two cores would be also very useful.

P9-L9 "(BP, referring to the10 drilling year of the respective ice core if not otherwise noted)".

This could be very confusing. I strongly suggest to use a different notation.

P10-L6 "The groups of peaks are separated by a comparatively stable signal of low Ca concentrations. The latter is interpreted as resulting from the varying degree of winter snow being included in the record otherwise dominated by summer snow. "

This "group of peaks" sounds very suspicious and arbitrary. This idea needs to be supported with additional evidences from snow pit studies or from a comparison with the seasonality of stable isotopes at depths where annual layers are still distinguishable.

P10-L8 "Accordingly, the grouped peaks correspond to sub-annual snow deposition events of elevated Ca concentration during the summer period".

How can multiple wet dust deposition events be distinguished from the formation of one or more summer surfaces of accumulated dust?

The different age depth relationships displayed in Fig. 5 for KCC and KCI between 150 years BP and 700-800 years BP needs to be carefully discussed. In fact, within this interval both cores do not depend on absolute time horizons.

Caption Fig. 4b The year to year correspondence between Ca annual layers determined by LA and CFA should be indicated drawing lines connecting the star symbols. At the moment no clear one-to-one link is apparent.

P12 L10 "The frequency of occurrence in these total snow loss events is, however, extremely hard to quantify. Counting annual layers in between the above mentioned (dust) horizons within the last century, reveals an offset of typically only one to two years as compared to the known age of the horizons."

The last century is not very much representative of colder periods (Little Ice Age, LIA) where snow drift could have been far more important, possibly eroding more annual layers. Notably the LIA is also the time when no absolute time horizons are available for the time scale that depends entirely on counting annual layers.

P13 L13 "(e.g. note the distinct isotope minima around 1360 AD)."

This is an important note when considering the companion paper by More et al. 2017 in Geohealth as this time corresponds almost exactly with the time of the Black Death. Could this isotope minima have been a large winter snow accumulation event? In this sense the implications for the interpretation of the linked Pb record could be very important.

P14 L14 "However, this is the first time that the correlation holds to this extent also for the comparatively old core sections".

As far as I know, this has been also observed at least in the Ortles ice cores (Gabrielli et al. The Cryosphere 10, 2779–2797, 2016).

P15 L5 "Stack of the two stable isotope records (calculated as their simple average)".

Please, mention the temporal step used to calculate averages.

P15 L15 "substantially higher sensitivity values for KCI than KCC, revealing 2.3 vs. 1.4 ‰âŮęC, respectively".

This is surprising considering the striking similarities of the two stable isotope profiles (Fig. 6). Could you provide an explanation?

P15 L16 "changes in snow preservation are expected to bias sensitivity (cf. the conceptual consideration in section 2.2)."

In this case, the conceptual model presented should be useful to quantify and perhaps correct this bias.

P15 L17 "This is consistent with the sensitivity difference among KCI and KCC, since an even more strict confinement towards sampling the high summer season can be expected for the lower accumulation KCI."

Agree, but from just a look of the two records, KCI and KCC show very similar absolute stable isotopes values during the potential calibration time (Fig. 6).

P17 L3 "While the particle signal alone is not sufficient for differentiating these events, Saharan dust layers in CG ice cores can be reliably identified based on the 5 analyses of Ca2+, supplemented by alkalinity measurements and, in principle, particle size distribution (Wagenbach et al., 1996)."

This information is not sufficient to discriminate between a Saharan dust event and a past summer surface formed by dry dust accumulation. While Saharan dust events may well have these characteristics, also summer dust layers formed during prolonged dry periods could have the same or similar characteristics. In addition, a higher coarse particle percentage may be more indicative of local dust rather than long-range transported dust.

P18 L5 "This is in broad agreement with periods of enhanced Saharan dust deposition reported by Thevenon et al. (2009) obtained from elemental analysis in a CG ice core."

This presumed broad agreement should be demonstrated in detail. At this time the high frequency of dust events in the described periods is both consistent with more frequent Saharan events and the formation of dust enriched past summer surfaces.

P19 L25 "The connection of the distinct increase in coarse particles with enhanced dust event frequency indicates an increase in direct transport of Saharan dust the (as opposed to indirect advection with longer pathway and thus stronger decrease in coarse particles)."

This is, at best, consistent (not indicative) with an increase in direct transport of Saharan dust. In fact this observation is also compatible with other scenarios (e.g. higher

occurrence of summer surfaces marked by dust, increase in the intensity of the summer vertical transport of dust of Alpine origin).

P19 L27 "Notably, this view is consistent with increased deuterium excess, which would be expected from warm and dry air masses collecting moisture over the Mediterranean."

This may be an important note that needs to be expanded and adequately referenced.

P20 L12 "increased meridional transport favoring direct Saharan dust advection over the Mediterranean is consistent with a NAO+ dominated MCA, lasting 1100 to about 1300 AD, as proposed by Trouet et al. (2009). Although NAO is mainly a winter signal (thus not a direct concern to CG summer representative ice core signals)"

As mentioned, Saharan dust advection is just one of the possibilities and a NAO signal consistent with this hypothesis is really a weak reasoning, especially considering that a NAO winter signal can have little impact on the summer biased cores from Colle Gnifetti. Please, consider to entirely remove this section (lines 10-17).

Conclusions: conclusions needs to be tuned down accordingly to the main observations performed.

P21-L12 "The intrinsic contribution of snow preservation may bias the isotope-temperature sensitivity".

This is interesting but it has not been shown and adequately discussed within the text.

---

## Referee Comment (RC2) · Anonymous Referee #2 · 25 Jul 2017

**Review of "*Temperature and mineral dust variability recorded in two low accumulation Alpine ice cores over the last millennium*" by Bohleber et al., 2017**

The paper presents an excellent dataset of stable water isotopes and other 'dust' proxies (i.e. insoluble particles and $Ca^{2+}$) from two separate ice cores drilled at Colle Gnifetti in the Pennine Alps, reaching back in time as far as a thousand year, a remarkable achievement for a European alpine ice core. This study combines a very good quality of data retrieval with a robust strategy regarding the dating, and therefore deserves to be published in Climate of the Past. The data treatment and statistical approach is also adequate and robust and only minor changes should be made. I will illustrate now few of the weaknesses that the manuscript presents and some suggestions on how to strengthen these points before the final publication. Detailed comments follow.

Firstly, the manuscript fails a bit in illustrating the reason why it is important to obtain a $Ca^{2+}$-derived temperature profile and what advantages/disadvantages this would have compared to a conventional $\delta^{18}O$-derived temperature profile. As mentioned in the abstract, the high and potentially non-stationary isotope/temperature sensitivity limits the quantitative use of the stable isotope ($\delta^{18}O$) variability and therefore a $Ca^{2+}$-derived temperature profile could provide essential information for a better constrain of temperature variability in the deepest (oldest) section of the two ice cores. This point should be highlighted more considering, however, that: i) $Ca^{2+}$ sensitivity to temperature changes might be, and it is likely to be, non-stationary as well over the last 1000 yrs; ii) the relationship between $Ca^{2+}$ and temperature could very well derive from post-depositional processes. This last point is particularly relevant (also considering that $NH_4$ show a similar temperature dependance) and the authors should elaborate more on why they think this is not the case. For example, if there is any data available of density, DEP or occurrence of melt layers, I suggest that the authors should use these data to back up some of their assumption regarding the summer-signal preservation by consolidation and its relationship with the seasonality of $Ca^{2+}$.

Furthermore, the assumption that the $Ca^{2+}$ signal is almost entirely expression of a dust input from Saharan region is not enough justified in the text. The fact that the $Ca^{2+}$ profile might derive from both wet and dry deposition and both proximal and distal sources cannot be ruled out from the data shown in the manuscript. Since the isotope/impurity co-variation on the inter-annual scale is mainly related to changes in the amount of winter precipitation contributing to annual mean values, I think is necessary to briefly consider different scenarios concerning the (although marginal) role of dry deposition in the Colle Gnifetti area and how these could change the $Ca^{2+}$ signal in the different cases.

While provenance studies (Sr and Nd isotopes for example) go beyond the scope of the work, I think a more detailed discussion on the comparison of the insoluble dust profile vs the $Ca^{2+}$ profile is necessary to utilize the calcium signal a proxy for Saharan dust input.

Whether Saharan_dust-$Ca^{2+}$ data is a reliable proxy for palaeotemperature is yet again another point that needs to be better illustrated in the text. I think the authors should provide more justification regarding why the $Ca^{2+}$ variability is mainly related to temperature changes and not, for instance, to changes at the dust source (Saharan desert).

**Detailed comments:**

Page 1 Line 1-2: I would update this statement in view of the recent 7000-yrs long ice core record from the Ortles (Gabrielli et al., 2017).

Page 3 Line 11-12: "which prevents any link of the climatologic precipitation rate to the net snow accumulation rate". I am not sure I understand here: Does this mean that the seasonality in the proxies is not governed by accumulation rate? Or is rather the longer-time variability? In any case I suggest changing the word "prevents" with "limits".

Page 3 Line 17: I found the wording a bit confusing. What "chemical/isotopic conditions" means? Do you mean chemical and isotopic signatures?

Page 4 Line 1-2: "the isotope/impurity co-variation on the inter-annual scale reflects to a large degree changes in the amount of winter precipitation contributing to annual mean values" I think is important here to highlight why the authors think dry deposition is playing a marginal role.

Page 4 line 10-11:"Therefore, the $Ca^{2+}$ record of the CG ice cores is primarily related to mineral dust and dominated by Saharan dust". It's hard to tell without provenance studies. I suggest using "dominated by dust, most likely originating in the Saharan desert".

Page 7 Line 3: "Deviations from a CPP of 50% indicate higher or lower contribution of large and small particles respectively". You have to exclude local sources of dust then if you want to use the threshold to distinguish Saharan dust layers. I would add a sentence justifying this.

Page 8 Line 1: I would specify what "Ca signal" means. Is it Intensity in counts per second? Or total counts? Please add this also to the relevant figures.

Page 9 Line 8: "Below 26 m WE the identification of annual layers became ambiguous and was abandoned". Maybe I missed this information, but why then LA-ICPMS was not performed on the KCI core? Please provide justification, if it is not provided somewhere else.

Page 13 Line 12: "due to the strong effect of isotope diffusion at CG, inter-annual or even seasonal isotope variability is effectively eliminated". What about $Ca^{2+}$ diffusion? While dust does not diffuse, the contribution of soluble particles to the $Ca^{44}$ signal should be briefly addressed too, together with their possible diffusion.

Page 14 Line 31-32: "From a preliminary inspection of snow pit data recently obtained for the KCI-KCC flow line, there is no clear indication of a systematic trend in mean $\delta^{18}O$ levels upstream of KCC, however." It might be worthy to consider adding a plot (at least in the supplementary material) showing this.

Page 15 Line 12: "higher sensitivity values for KCI than KCC, revealing 2.3 vs. 1.4 ‰/∘C, respectively".

This discrepancy seems surprisingly high even considering the difference in accumulation rate that you correctly highlight. Could it be related also to the strong isotope diffusion at CG?

Page 20 Line 10-17: This entire section seems a bit far-fetched. As the authors said, the summer-bias signal at CG strongly advocate against a NAO imprint on the KCC and KCI temperature reconstruction. I suggest adding few more considerations to justify this link or remove the entire section.

Page 21 Line 1-20: I suggest to the authors to add a sentence outlining the feasibility of using $Ca^{2+}$ records for temperature reconstruction in other alpine site, or generally in other low accumulation ice core site.

**References**

Gabrielli, P., Barbante, C., Bertagna, G., Bertó, M., Carturan, L., Dinale, R., & Seppi, R. (2017, April). 7000 year European climate record from the Ortles ice core. In EGU General Assembly Conference Abstracts (Vol. 19, p. 9932).

---

## Author Comment (AC1) · 30 Sep 2017

The response has been uploaded as a supplement.

Please also note the supplement to this comment:
https://www.clim-past-discuss.net/cp-2017-80/cp-2017-80-AC1-supplement.zip

———————————————————

---

## Author Comment (AC2) · 30 Sep 2017

The response has been uploaded as a supplement.

Please also note the supplement to this comment:
https://www.clim-past-discuss.net/cp-2017-80/cp-2017-80-AC2-supplement.zip

———————————————————

---

## Author Response (AR1)

| 1  | "Temperature and mineral dust variability recorded in two low accumulation Alpine           |
|----|---------------------------------------------------------------------------------------------|
| 2  | ice cores over the last millennium" by Pascal Bohleber et al.                               |
| 3  | - Response to reviews -                                                                     |
| 4  |                                                                                             |
| 5  | Please note:                                                                                |
| 6  | • All line numbers in "Changes to manuscript" refer to the new version (if not              |
| 7  | noted otherwise)                                                                            |
| 8  | • Changes in the corresponding pdf are highlighted in red                                   |
| 9  | • Author's responses to the referee's comments are in blue                                  |
| 10 | • All new references can be found in the new manuscript                                     |
| 11 |                                                                                             |
| 12 | Introductory remark:                                                                        |
| 13 | We thank both referees for their very thorough reviews and we appreciate the helpful        |
| 14 | suggestions and comments. After careful consideration, especially of points commonly        |
| 15 | raised by both reviewers, we determined the need to clarify our basic line of argument. For |
| 16 | this purpose, we would like to emphasize the following key points:                          |
| 17 | 1) We aim to distinguish throughout the paper two separate signal components of the         |
| 18 | Ca2+ record: 1. Episodic spikes , typically two orders of magnitude above            |
| 19 | background levels, and 2. Long-term trends of the decadal-scale average Ca2+         |
| 20 | concentration. Both components are evaluated separately. At CG, mineral                     |
| 21 | background aerosol levels are generally low and the Ca2+ record is dominated by             |
| 22 | inputs of Saharan dust (e.g. Wagenbach et al. 1996). In this sense, the already             |
| 23 | established link between Ca2+ and Saharan dust concerns signal component 1. The             |
| 24 | potential new link between Ca2+ and temperature is evaluated for signal                     |
| 25 | component 2.                                                                                |
| 26 | 2) Regarding 1., we do not intend to make quantitative inferences regarding mineral         |
| 27 | dust concentrations of individual events but aim to estimate their frequency of             |
| 28 | occurrence at CG. For this purpose we build on what has already been demonstrated           |
| 29 | in previous studies, namely that Ca2+ combined with an alkalinity measure is in fact        |
| 30 | a sensitive and appropriate tool to identify Saharan dust layers at CG (Wagenbach et        |
| 31 | al., 1996).                                                                                 |
| 32 | 3) Regarding 2., we respond to the intriguing present situation at CG where we face i)      |
| 33 | fundamental shortcomings in making quantitative use of the stable water isotope             |

| 34 |         | thermometer (Bohleber et al. 2013) and ii) the already known co-variation between          |
|----|---------|--------------------------------------------------------------------------------------------|
| 35 |         | trends in Ca2+ and delta O-18 (Wagenbach et al. 1996, Wagenbach and Geis, 1989)            |
| 36 |         | as well as delta 0-18 and instrumental temperature (Bohleber et al. 2013). This            |
| 37 |         | raises the question to what extent a relationship exists between temperature and           |
| 38 |         | Ca2+ trends, and if this may serve as a potential substitute for quantitative              |
| 39 |         | temperature reconstruction at CG.                                                          |
| 40 | 4)      | While we explore the suggested relation between Ca2+ trends and temperature, we            |
| 41 |         | strongly emphasize that it is not our intention to introduce a new ice core                |
| 42 |         | temperature proxy. We evaluate the Ca2+ trends solely regarding their site-specific |
| 43 |         | temperature connection. This is an analogue approach as pursued for NH4+ in the            |
| 44 |         | Bolivian Andes (Kellerhals et al. 2010).                                                   |
| 45 | 5)      | We also emphasize that we by no means disregard the influence of snow deposition           |
| 46 |         | and post-depositional effects. In fact, the main goal in using the semi-quantitative       |
| 47 |         | snow deposition model (section 2.2) is to demonstrate that post-depositional               |
| 48 |         | influence may not be disregarded when evaluating the temperature coupling to               |
| 49 |         | Ca2+-trends.                                                                               |
| 50 | In orde | er to eliminate the apparent ambiguities in the original version and in order to make      |
| 51 | our lin | e of argument more clear we have made the following major changes to the                   |
| 52 | manus   | cript. We feel that by means of these changes the most important issues raised by the      |
| 53 | review  | ers have been properly addressed and the clarity of the paper has been substantially       |
| 54 | improv  | red. Detailed responses to the referees' comments are given separately for referee #1      |
| 55 | and #2  | , respectively.                                                                            |
| 56 | Chang   | es to manuscript:                                                                          |
| 57 | •       | We have clarified the abstract and the conclusions according the above points. We          |
| 58 |         | now present two additional tables and two additional figures as supporting                 |
| 59 |         | evidence in the appendix / as supplementary material.                                      |
| 60 | •       | Page 4 Lines 29ff.: We added a clear statement regarding the separate treatment of         |
| 61 |         | Ca2+-spikes and long-term variability in this study.                                       |
| 62 | •       | We have split up the previous section 2.2. as follows:                                     |
| 63 |         | • Page 3 Lines 16ff.: We combined with the original section 2.1 the                        |
| 64 |         | fundamental description of snow preservation at CG and its consequences                    |
| 65 |         | for interpreting the isotope and mineral dust proxies. This also includes the              |

| 66 | basic reasoning for expecting a temperature-related imprint in the long-                    |
|----|---------------------------------------------------------------------------------------------|
| 67 | term Ca2+-variability.                                                                      |
| 68 | • Since we feel like it has diverted the attention from our main line of                    |
| 69 | argument, we have moved the details of the semi-quantitative treatment of                   |
| 70 | snow deposition to the supplementary material in the appendix.                              |
| 71 | • Page 18, Line 4ff.: We now refer to the semi-quantitative analysis at a later             |
| 72 | point in the manuscript. The discussion of potential causes of the observed                 |
| 73 | Ca2+-temperature co-variation is now presented within 5. Results and                        |
| 74 | Discussion. We believe this makes it easier to follow for the reader, since the             |
| 75 | results have been presented at that point.                                                  |
| 76 | • Page 18, Line 21ff: We have included a clear statement regarding the site-specific        |
| 77 | nature of the observed Ca2+-temperature connection.                                         |
| 78 |                                                                                             |
| 79 | Response to anonymous referee #1                                                            |
| 80 | This paper presents an exceptional data set composed by stable isotopes and various         |
| 81 | proxies of aeolian dust deposition as measured in two Alpine cores from Colle Gnifetti      |
| 82 | (European Alps) during the last 1000 years. The quality of the data presented appears       |
| 83 | robust and, in general, the time scale adopted seems reliable. However, while this data set |
| 84 | does deserve publication, the interpretation and use of the results obtained is often weak, |
| 85 | speculative and potentially misleading. I believe the discussion section of the journal     |
| 86 | Climate of the Past represents a suitable venue to possibly sort out several important      |
| 87 | interpretative issues before the possible final publication.                                |
| 88 | We thank the referee for the detailed comments. As outlined in the introductory remark, we  |
| 89 | feel that there is an apparent ambiguity regarding the interpretation of the results, which |
| 90 | we have now tried to remove and believe that by this means the main issues raised by the    |
| 91 | reviewer can be addressed properly.                                                         |
| 92 |                                                                                             |
| 93 | Regardless the source of the dust deposited, the correlation between Ca2+ (as dust proxy)   |
| 94 | and atmospheric temperature may be really linked to a poorly known post depositional        |
| 95 | process. Dust in surface snow may indeed facilitate a metamorphic process that could        |
| 96 | consolidate and conserve the snow on site, for instance via a reduced albedo (larger        |
| 97 | amounts of solar radiation absorbed) at times with more frequent clear sky and higher       |
| 98 | atmospheric temperatures. In my view the authors should really try to better describe this  |

99 possible process.

| 100 | We fully agree that a post-depositional influence must be taken into account, and the        |
|-----|----------------------------------------------------------------------------------------------|
| 101 | mechanism described by the reviewer is entirely consistent with our general view             |
| 102 | regarding the post-depositional effect of dust-rich surface snow. We have added a more       |
| 103 | detailed description of this process. In addition we now provide supporting evidence for the |
| 104 | influence of dust on snow consolidation. We have included a comparison between Ca2+ and      |
| 105 | the high-resolution density profile to show directly that in general dust-rich snow layers   |
| 106 | correspond with confined layers of enhanced density (with respect to the surrounding         |
| 107 | layers). This supports the concept of dust-rich layers featuring faster snow consolidation   |
| 108 | (Figure A2).                                                                                 |
| 109 | Changes to manuscript:                                                                       |
| 110 | • Changes according to splitting the original section 2.2 as outlined in the initial         |
| 111 | remarks (see above).                                                                         |
| 112 | • Page 18, Line 4ff.: We have included in this discussion our supporting evidence from       |
| 113 | the high-resolution density data. We also now include references for each potential          |
| 114 | driver for the Ca2+-temperature co-variation.                                                |
| 115 |                                                                                              |
| 116 | Considering the poor seasonality of intense dust events of Saharan origin, at this time I do |
| 117 | not see how their semi-quantitative model can be useful to support the possible occurrence   |
| 118 | of out of phase dust (wet) deposition and post-depositional consolidation. Most              |
| 119 | importantly, the out of phase dust/snow deposition and the possible post depositional        |
| 120 | process linked to atmospheric temperature is not sufficiently supported by data. In this     |
| 121 | respect results from past snow pit studies might be illuminating and should be extensively   |
| 122 | presented.                                                                                   |
| 123 | As outlined in the introductory remark, the main purpose of the semi-quantitative model is   |
| 124 | to demonstrate that snow deposition can have a non-negligible influence on the long-term     |
| 125 | variability of Ca2+ and thus must be considered as a potential contribution to the Ca2+-     |
| 126 | temperature coupling. We have clarified this view accordingly and now also include           |
| 127 | additional information in the text regarding the following points:                           |
| 128 | • Snow pit data presented in earlier studies (Wagenbach and Geis, 1989) show that            |
| 129 | local maxima of stable isotope and dust-proxies coincide. This is consistent with            |
| 130 | what we find in the uppermost section of our core, providing a near-seasonal                 |
| 131 | resolution (Figure A1). Analogue to the treatment of the stable isotope signal in            |

| 132 | Wagenbach et al. (2012), it is thus justified to assume a potential phase-lag between          |
|-----|------------------------------------------------------------------------------------------------|
| 133 | accumulation and Ca2+ maxima. We also note that the principal function of the                  |
| 134 | phase lag is to mimic the seasonally biased sub-sampling. Qualitatively similar                |
| 135 | results are found when using a different approach to model this process, e.g. by               |
| 136 | following Fisher and Koerner (1988).                                                           |
| 137 | Changes to manuscript: Include Figure A1 as additional supporting evidence in the              |
| 138 | appendix / supplementary material                                                              |
| 139 | • The connection between atmospheric temperature and snow consolidation is well                |
| 140 | established, e.g. temperature being used as parameter in various firn-densification            |
| 141 | models (also consider Figure 27 in Fauve et al. (2002)). A faster snow consolidation           |
| 142 | driven by higher summer temperature corresponds in the model to an increase in                 |
| 143 | a_r and a decrease in t_phi. We have clarified this accordingly.                               |
| 144 | Changes to manuscript: Added respective references to discussion on page 18, lines             |
| 145 | 4ff. Added text and clarified the influence of temperature via snow consolidation in the       |
| 146 | semi-quantitative model (appendix A1).                                                         |
| 147 | • We also now explicitly point out that the seasonality in Ca2+ is not dominated by the        |
| 148 | seasonal occurrence in Saharan dust events, but rather the result of increased                 |
| 149 | vertical mixing of air masses in summer vs. overall clean winter snow conditions (cf.          |
| 150 | Figure A2).                                                                                    |
| 151 | Changes to manuscript: Added text and references on page 4, lines 17ff.                        |
| 152 |                                                                                                |
| 153 | While the occurrence of a Saharan dust fallout via wet deposition is very well established at  |
| 154 | Colle Gnifetti, the interannual variability of this kind of Saharan events is very high in the |
| 155 | Alps and is mostly related to periods when atmospheric advection dominates (late fall,         |
| 156 | winter, early spring). During summer, however, atmospheric vertical convection rules in the    |
| 157 | Alps and dry transport and deposition of dust of local/Alpine origin is extremely likely and   |
| 158 | probably intense also on Colle Gnifetti. Source reconstruction of dust entrapped in ice cores  |
| 159 | is very complex even when sophisticated multiple proxies are used (e.g Sr and Nd isotopes).    |
| 160 | Ca2+, even when combined with ECM and dust size, cannot be considered a specific proxy of      |
| 161 | Saharan dust events and cannot discriminate with sufficient confidence between different       |
| 162 | kinds of sources (e.g. Saharan vs. local) and atmospheric circulation (e.g. meridional flows   |
| 163 | vs. vertical convection).                                                                      |
|     |                                                                                                |

164 We agree that vertical atmospheric convection likely rules the seasonal cycle of Ca2+. The

165 sophisticated isotopic fingerprinting of individual dust layers is certainly a worthwhile 166 future target (and we appreciate the suggestion), although requiring substantial analytical 167 capabilities as well as a significant portion of the ice core. For our main target being 168 quantifying the frequency of occurrence of Saharan dust events, the task was to identify the 169 specific imprint of Saharan dust in presence of events of increased impurity load by 170 enhanced vertical mixing. Here we were able to rely on what has already been shown in the 171 two earlier studies investigating dust-related records at CG (i.e. Wagenbach et al. (1996) 172 and Wagenbach and Geis (1988)). These studies have clearly demonstrated that 173 investigating "whether the size distribution parameters of the Saharan dust • 174 deposited differ significantly from those of dust deposits related to more regional 175 source areas or even background aerosol" revealed that "the volume size 176 distribution of the two Saharan dust events is indeed significantly shifted towards 177 larger particles" (Wagenbach and Geis, 1988, and cf. Figure 7 therein). The authors 178 also conclude that "it seems reasonable to attribute individual differences in the size 179 distribution of the Saharan dust deposited on CG to different transport times which 180 in turn means different transport velocities and/or different lengths of the 181 trajectories". 182 "that the Ca2+ spikes associated with the strongly alkaline snow layers in recent firn • 183 at Colle Gnifetti are most likely due to the mobilisation of calcerous Saharan soil 184 material which, following long range transport to the Alps, is mainly deposited there 185 by precipitation scavenging" (Wagenbach et al. 1996). The authors compile data 186 from various CG snow pits and ice cores, finding that "This compilation suggests that 187 the analyses of ionic Ca2+ in connection with reliable alkalinity measurements 188 appear to be a very sensitive and specific tool to identify Saharan dust influenced 189 snow layers in high depth resolution". 190 We now make additional clear reference to these specific earlier findings to point out that it 191 is not the scope of our work to present a refined detection of Saharan dust in CG ice cores. 192 Instead, we intend to make use of already established tools, in combination with our new 193 ice core chronology comprising the last millennium at sufficient confidence. 194 Changes to manuscript: Page 4, Lines 26-28: Added text and reference accordingly. 195 196 Remarkably, if Ca2+ concentration is assumed to really trace dust from a specific area (e.g. 197 Sahara desert) this would severely prevent its general use as paleothermometer as this

| 198 | parameter would strongly depend on environmental conditions at the source that are not       |
|-----|----------------------------------------------------------------------------------------------|
| 199 | directly related to atmospheric temperature. For instance a Ca2+ based paleo thermometer     |
| 200 | might be severely biased by different soil conditions at the source, due for instance to     |
| 201 | changes in atmospheric circulation and precipitation, not temperature (e.g. green Sahara     |
| 202 | during the middle-early Holocene). The use of Ca2+ to construct a paleothermometer could     |
| 203 | thus depend on very site specific post-depositional processes and on changing                |
| 204 | environmental conditions at the source of dust over time. Thus it is fundamental to greatly  |
| 205 | caution about its general use in space (drilling site dependence) and time (it may work only |
| 206 | during certain times). In conclusion, while the relation between Ca2+ and temperature is     |
| 207 | interesting, its extended use cannot provide unambiguous novel knowledge about past          |
| 208 | atmospheric temperature.                                                                     |
| 209 | Here we refer again to our initial comment regarding the separate treatment of Ca2+ spikes   |
| 210 | (Saharan dust proxy together with an alkalinity measure) and the overall trend component     |
| 211 | of the Ca2+ signal. Only the latter is explored with respect to a potential coupling to      |
| 212 | temperature, and we are now being more explicit regarding this distinction. We are also      |
| 213 | clearly stating that, at best, Ca2+ is a site-specific temperature proxy. Regarding its      |
| 214 | performance to reconstruct temperature in time, please see the next comment below.           |
| 215 | Changes to manuscript:                                                                       |
| 216 | • Page 4, Lines 29ff.: Added statement regarding distinction of signal components in         |
| 217 | Ca2+.                                                                                        |
| 218 | • Page 18, Lines 21ff.: Added statement regarding site-specific role of Ca2+.                |
| 219 |                                                                                              |
| 220 | When compared to Ca2+, the use of stable isotopes as a proxy of atmospheric temperature      |
| 221 | is better justified by the physics and is less sensitive to source effects due to moisture   |
| 222 | changes. It is remarkable that the correlation of stable isotopes and instrumental           |
| 223 | atmospheric temperature is very good during the last 150 years when the recorded             |
| 224 | instrumental temperatures are most robust. It appears that the real problem of the possible  |
| 225 | paleo-thermometer based on stable isotopes is the "excessive" and spatially variable         |
| 226 | sensitivity (1.4-2.3 per mill/C) when compared to what one would expect (0.65 per mill/C).   |
| 227 | For this reason the authors decide not to attempt a quantitative temperature reconstruction  |
| 228 | based on stable isotopes. However, while this decision may be justified, I believe that an   |
| 229 | extended discussion of the possibly biased paleo-temperatures obtained by means of this      |
| 230 | presumably too sensitive/variable paleo-thermometer would be very interesting. This          |

231 could show for instance the inconsistency of the temperatures obtained, even considering

physical processes such as the amplification of atmospheric temperature anomalies with

the elevation.

234 We believe that an adequate discussion of the lags and leads of the peculiar high

- isotope/temperature sensitivity is beyond the scope of this work and would likely result in
- an entirely different paper. Ideally, such an evaluation would need to take into account
- regional isotope-climate modeling. We take the reviewers comment as an encouragement to
- 238 pursue such a dedicated assessment in a future study. The agreement between the isotope
- and temperature trends within the last 150 years has been discussed in an earlier study,
- $240 \qquad also showing that if calibrated with the last 100 years, isotope levels at CG suggest a warmer$
- 241 "early instrumental period" by about 0.4 deg C (Bohleber et al. 2013). Following this
- 242 comment we now include this point in our discussion.
- 243 Remarkably, however, if considering the Ca2+ trends against instrumental temperature, we
- find an agreement of similar quality as for the isotopes, but now persistent over the full
- instrumental 250-year period (cf. Figure 7). Although the exact reason for a potential non-
- stationary isotope-temperature sensitivity remains elusive, this finding suggests that the
- 247 potential Ca2+-temperature relationship may not be affected to the same degree. Moreover,
- $248 \qquad \text{if using the Ca2+-temperature for a tentative calibration, the reconstructed temperature} \\$
- 249 variability is consistent with the latest summer temperature reconstruction based on other
- archives (the Luterbacher et al. (2016) record). The main point here is that given the
- agreement with instrumental data and other temperature reconstructions, we find no
- evidence of potential non-stationary behavior in the Ca2+-temperature correspondence.
- 253 We have added text in order to be more clear about this interpretation.
- 254 Changes to manuscript: Page 19, Lines 13ff.: Added text to clarify the interpretation of the255 comparison with the other proxy reconstruction.
- 256
- 257 A likely reliability of the timescale obtained for the two cores from Colle Gnifetti is
- suggested by the good correlation of Ca2+ with an independent, well dated, past summer
- temperature record obtained from tree rings. However, the reliability of this time scale may
- be just due to the use of recent absolute time horizons (during the last century) and 14C
- 261 measurements (although the significant reference, PhD thesis of Hoffmann 2016, cannot
- 262 provide an accessible and peer reviewed methodological support). Counting of annual
- 263 layers is not convincing in the deep part of core KCC. In particular the concept of "group of

264 peaks" of Ca2+ to identify a single annual layer seems extremely arbitrary, and, at this time, 265 not supported in the paper by any snow pit observation. In addition a possibly larger 266 deflation of lighter snow during the colder summers of the Little Ice Age may have removed 267 more annual layers that expected. In this way, it is very possible that the interval period 268 between 150 and 600 BP remains unconstrained and prone to larger uncertainties. 269 We do not use the comparison with the summer temperature reconstruction to show the 270 reliability of our time scale. This would require comparing an already demonstrated 271 temperature signal reconstructed from the ice core. Our argument goes in the opposite 272 direction: Having constructed a reliable time scale, we look for consistency between the 273 potential Ca2+-based temperature variability and an established reconstruction from 274 another archive. 275 We believe we have demonstrated the reliability of our time scale for the following reasons: 276 Annual layer counting is a well-established and widely used tool for ice core dating. 277 However, at CG the employment of this tool was so far limited by rapid layer thinning 278 beyond the resolution of most melting techniques. Based on the agreement between CFA 279 and LA-ICP-MS, we were able to make extended use of annual layer counting to build our 280 chronology. While the identification of additional absolute dating horizons, such as volcanic 281 eruptions, would of course be desirable, this is at present not possible for CG. However, we 282 have taken great care to account for the resulting uncertainty in "unconstrained" annual 283 laver counting (section 4.4). The resulting age-scale is backed by 14C markers, an approach 284 already employed successfully at CG in previous studies (e.g. Jenk et al. 2009). The PhD 285 thesis by Helene Hoffmann has been reviewed as part of the process in obtaining her PhD at 286 Heidelberg University. It is fully available online and we have now included the link in the 287 respective reference. In addition, we are now able to reference her according publication 288 which has been accepted for publication in *Radiocarbon* in the meantime (Hoffmann et al. 289 2017). 290 We thank the referee for bringing our attention to provide additional justification for the 291 concept of "grouped peaks" in the LA-ICP-MS Ca signal. For this purpose we have prepared 292 a supplementary figure showing the uppermost 5 m of the delta O-18 and Ca2+ stratigraphy 293 in our core. Here, multiple sub-seasonal peaks in Ca2+ are clearly present to back up this 294 concept. In addition, we again refer to the fact that based on its ultra-high resolution and 295 non-destructive technique, LA-ICP-MS can map the spatial heterogenity of impurities even

within a single annual layer.

| 297 | Changes to manuscript:                                                                         |
|-----|------------------------------------------------------------------------------------------------|
| 298 | • Page 19, Lines 13ff.: Clarified regarding the comparison with the other proxy                |
| 299 | reconstruction not being intended for dating verification.                                     |
| 300 | • Added reference for Hoffmann et al. (2017a,b)                                                |
| 301 | Added Figure A1 as supporting evidence for "grouped peak" concept                              |
| 302 |                                                                                                |
| 303 | While Colle Gnifetti is a very well know ice core drilling site and many glaciological studies |
| 304 | have been performed, this paper fails to offer the necessary comparison with existing data     |
| 305 | set available from other ice cores (and snow pit samples) obtained at the same site. In        |
| 306 | particular several studies of particulate and aerosol deposition were performed (e.g.          |
| 307 | Thevenon, JGR 2009) and should be carefully compared to check the consistency (or not) of      |
| 308 | the new findings with the previous results.                                                    |
| 309 | We already compare our results to the findings of Thevenon et al. (2009) on page 18 line 5     |
| 310 | (original manuscript). However, a more detailed comparison is hampered given the               |
| 311 | difference in depth resolution and dating uncertainty between the two studies. We would        |
| 312 | certainly welcome a specific suggestion by the referee regarding the further comparison of     |
| 313 | our results.                                                                                   |
| 314 | Changes to manuscript: Page 17, Lines 7ff.: Added some more detail regarding the               |
| 315 | comparison with Thevenon et al. (2009)                                                         |
| 316 |                                                                                                |
| 317 |                                                                                                |
| 318 | Specific comments:                                                                             |
| 319 |                                                                                                |
| 320 | P1 L1: "Among ice core drilling sites in the European Alps, the Colle Gnifetti (CG) glacier    |
| 321 | saddle is the only one to offer climate records back to at least 1000 years"                   |
| 322 |                                                                                                |
| 323 | There is now in the Eastern Alps a new ice core climate record that goes back almost 7000      |
| 324 | years (Gabrielli et al. The Cryosphere 10, 2779–2797, 2016; Gabrielli et al. 19th EGU          |
| 325 | General Assembly, EGU2017, proceedings from the conference held 23-28 April, 2017 in           |
| 326 | Vienna, Austria., p.9932).                                                                     |
| 327 | Thank you for pointing this out. We have changed our wording in order to be more specific.     |
| 328 | Colle Gnifetti is the only non-temperate site (Ortles in the Eastern Alps is partially         |
| 000 |                                                                                                |

329 temperate).

| 330 | Changes to manuscript: Page 1, Line 1 and Page 2, Line 5: Changed wording accordingly.   |
|-----|-------------------------------------------------------------------------------------------------|
| 331 |                                                                                                 |
| 332 | P1 L8: "A high and potentially non-stationary isotope/temperature sensitivity limits the        |
| 333 | quantitative use of the stable isotope variability thus far".                                   |
| 334 |                                                                                                 |
| 335 | This statement is not discussed sufficiently within the text.                                   |
| 336 | In this case the statement primarily serves as background and motivation (referring to what     |
| 337 | is already known at CG).                                                                        |
| 338 | Changes to manuscript: Page 1, Lines 7ff.: We have changed the text accordingly to make         |
| 339 | this more clear.                                                                                |
| 340 |                                                                                                 |
| 341 | P1 L15: "the medieval climate period around 1100–1200 AD stands out through an                  |
| 342 | increased occurrence of dust events, potentially resulting from a relative increase in          |
| 343 | meridional flow and dry conditions over the Mediterranean".                                     |
| 344 |                                                                                                 |
| 345 | While the frequency of the dust horizons is reproducible in the two cores, they cannot be       |
| 346 | linked to individual dust events of Saharan origin that cannot be unambiguously                 |
| 347 | distinguished from the occurrence of local past summer surfaces marked by dust                  |
| 348 | accumulation.                                                                                   |
| 349 | As pointed out in our response above, we are making use of a tool developed in an earlier       |
| 350 | study and demonstrated to be suitable to identify Saharan dust events in CG ice cores           |
| 351 | (Wagenbach et al. 1996). Following the discussion with the referee, however, we now             |
| 352 | include a statement that more sophisticated methods based on isotopic fingerprinting exist      |
| 353 | today and may be used in the future to test and refine our findings.                            |
| 354 | Changes to manuscript: Page 20, Lines 15ff. Added text with a respective statement.             |
| 355 |                                                                                                 |
| 356 | P2 L5: "Colle Gnifetti (CG) in spite of its limited glacier depth – stands out as the only site |
| 357 | where net snow accumulation is low enough to provide records over the last millennium           |
| 358 | and potentially beyond at a reasonable time resolution".                                        |
| 359 |                                                                                                 |
| 360 | Again, this is not correct, please see Gabrielli et al. The Cryosphere 10, 2779–2797,           |
| 361 | 2016.                                                                                           |
| 362 | Thanks- we changed our wording accordingly (see above).                                         |

| 364 | P4 L5 "A single deposition event typically lasts less than a few days (Sodemann et al., 2005; |
|-----|-----------------------------------------------------------------------------------------------|
| 365 | Schwikowski et al., 1995). The associated warm air temperature and the substantially          |
| 366 | lowered snow albedo both support surface snow consolidation and partly protect the dust       |
| 367 | layer from wind erosion."                                                                     |
| 368 |                                                                                               |
| 369 | As long as air temperature is below the freezing level (as during snow events), this cannot   |
| 370 | be a factor facilitating snow consolidation.                                                  |
| 371 | As discussed in our response above, temperature does play an important role in snow           |
| 372 | consolidation.                                                                                |
| 373 | Changes to manuscript: Page 18, Lines 4ff.: We have added a specific remark and an            |
| 374 | according references regarding this point.                                                    |
| 375 |                                                                                               |
| 376 | P4-10 "Therefore, the Ca2+ record of the CG ice cores is primarily related to mineral         |
| 377 | dust and dominated by Saharan dust related spikes".                                           |
| 378 |                                                                                               |
| 379 | This conclusion is unsupported by more recent data detailing more specific proxies of         |
| 380 | Saharan dust.                                                                                 |
| 381 | As indicated in the text, this statement refers to findings already published by two previous |
| 382 | studies (Wagenbach et al. 1996, Wagenbach and Geis 1988). If the referee would like to        |
| 383 | suggest a specific study that is providing new (in particular refuting) evidence, we would    |
| 384 | certainly consider this in our discussion.                                                    |
| 385 |                                                                                               |
| 386 | P4-27 "For instance, warm summers feature increased vertical mixing and hence a higher        |
| 387 | atmospheric impurity load, and in addition, entail faster fresh snow consolidation. This may  |
| 388 | lead to an increased relative amount of impurity-rich summer snow deposition."                |
| 389 |                                                                                               |
| 390 | This is a very reasonable and, unlike the meridional Saharan advection, a more regular        |
| 391 | process in the Alps. I'm not sure why the authors do not consider and discuss it further      |
| 392 | within the text.                                                                              |
| 393 | From the discussion with the referee we learned that we have not emphasized this point        |
| 394 | strongly enough, although we believe it is a very important process in this context. In fact  |
| 395 | part of this process is what we explore with the semi-quantitative model exercise. We have    |

| clarified this and come back to it in the respective parts of the Discussion.                      |
|----------------------------------------------------------------------------------------------------|
| Changes to manuscript:                                                                             |
| • Page 3, Lines 18ff.: Clarified and added text.                                                   |
| • Page 18, Lines 4ff.: Included a detailed discussion of all involved processes.                   |
|                                                                                                    |
| P5-L7 "Here we follow the model of Wagenbach et al. (2012), which assumes sinusoidal               |
| cycles for the precipitation-borne signal $S(t)$ and the surface accumulation pattern $A(t)$ , and |
| a phase-lag t' between S(t) and A(t)."                                                             |
|                                                                                                    |
| While accumulation and delta 180 seasonal patterns are accepted at Colle Gnifetti, it is           |
| much less so for a disturbed seasonal signal like dust (Ca2+). The authors need to present         |
| data supporting how the sinusoidal assumption is justified for Ca+2 deposition.                    |
|                                                                                                    |
| Paragraph 2.2 In general I do not find that this paragraph well written or even necessary. It      |
| is not clearly explained what are the main motivations and conclusions of the conceptual           |
| model. At this time I'm also not sure how it supports the rational of the interpretation. Does     |
| the model show that a phase lag between deposition and consolidation explain some recent           |
| observations? If so, the phenomenology of these processes needs to be supported,                   |
| displaying existing data (e.g. snow pits) that could complement the conceptual discussion          |
| performed by means of this model.                                                                  |
| As outlined in the initial comment and after careful consideration of the referee's comment        |
| we have decided to break up the original paragraph as not to divert from our main line of          |
| argument. Being now used in the Discussion the scope of the model consideration becomes            |
| more clear, i.e. to semi-quantitatively demonstrate that the influence of snow preservation        |
| may not be disregarded when evaluating the long-term variability of Ca2+ and that it must          |
| be considered as a potential process introducing the coupling to temperature.                      |
| Albeit the sinusoidal pattern is of course an idealization, we have now included additional        |
| evidence for the seasonal pattern and link between delta O-18 and Ca2+ (Figure A1).                |
| Changes to manuscript:                                                                             |
| • Page 18, Lines 4ff.: Moved and rewrote this section (originally part of 2.2) to clarify          |
| • Additional supporting figures in the appendix (Figure A1, A2)                                    |
|                                                                                                    |
| P7-L2 "The threshold (4.0 $\mu m$ ) was chosen such that it corresponds to the expected median     |
|                                                                                                    |

429 particle diameter of Saharan dust particles at CG (Wagenbach and Geis, 1989)."

430 431 Is this threshold necessary AND sufficient to discriminate between Saharan and non-432 Saharan dust? I do not think so. The reference reported is pretty old and in the meantime 433 many tools have been developed to characterize dust sources. In my view this threshold is 434 just indicative but not strictly discriminant. 435 Although the reference is old we are not aware that its findings have been refuted by newer 436 studies. As said before, we certainly agree that more sophisticated tools may potentially 437 allow for a precise fingerprinting of the individual dust sources, although requiring 438 substantial analytical effort. 439 We intended to use the already established discrimination method based on the median 440 particle diameter and set our threshold according to the previous study. That said, the 441 results are qualitatively independent with respect to the exact choice of the threshold. 442 Notably this includes the outstanding feature in CPP during the medieval period. 443 Only the sensitivity of the CPP to changes in the particle size distribution (PSD) is 444 dependent on the threshold. Choosing the threshold to be the median value of the normal 445 PSD means that the CPP is 50% for "regular" dust and is sensitive to changes in the PSD. 446 Changes to manuscript: Page 20, Line 7-8: We added a statement to point out that the 447 outstanding feature during the medieval period is not a result of the choice of threshold. 448 449 3.2.1 Radiocarbon analysis. A table detailing all the results obtained by analyzing the 6+5 450 samples from KCC and KCI needs to be reported, including the linked uncertainties. 451 We have added a table as requested and would also like to point out that we were able to 452 include an additional radiocarbon measurement for KCI, in clear support of the age scale. 453 **Changes to manuscript:** We included a respective table in the appendix, Table A2. 454 455 4 Ice core dating. A Table summarizing all the time horizons used in KCC and KCI needs to 456 be reported. Another table indicating different kind of annual layer counting in different 457 sections of the two cores would be also very useful. 458 **Changes to manuscript:** We included a respective table in the appendix, Table A1. 459 460 P9-L9 "(BP, referring to the drilling year of the respective ice core if not otherwise noted)". 461

| 462 | This could be very confusing. I strongly suggest to use a different notation.               |
|-----|---------------------------------------------------------------------------------------------|
| 463 | We have generally changed this notation to either year AD or stating the precise year, e.g. |
| 464 | year b2005 or year b2013 in order to be more precise.                                       |
| 465 |                                                                                             |
| 466 | P10-L6 "The groups of peaks are separated by a comparatively stable signal of low Ca        |
| 467 | concentrations. The latter is interpreted as resulting from the varying degree of winter    |
| 468 | snow being included in the record otherwise dominated by summer snow. "                     |
| 469 |                                                                                             |
| 470 | This "group of peaks" sounds very suspicious and arbitrary. This idea needs to be supported |
| 471 | with additional evidences from snow pit studies or from a comparison with the seasonality   |
| 472 | of stable isotopes at depths where annual layers are still distinguishable.                 |
| 473 | We thank the reviewer for bringing this to our attention. As requested we provide           |
| 474 | additional support of this concept in a supplementary Figure A1.                            |
| 475 |                                                                                             |
| 476 | P10-L8 "Accordingly, the grouped peaks correspond to sub-annual snow deposition events      |
| 477 | of elevated Ca concentration during the summer period".                                     |
| 478 |                                                                                             |
| 479 | How can multiple wet dust deposition events be distinguished from the formation of one or   |
| 480 | more summer surfaces of accumulated dust?                                                   |
| 481 | We do not see how, based on the LA-ICP-MS Ca signal alone, one could distinguish wet and    |
| 482 | dry deposition. However, the comparison with the CFA-based counting in depth intervals      |
| 483 | where CFA clearly identifies the annual layer signal (Figure 4 b)) shows that the "grouped  |
| 484 | peaks" are not a result of multiple, very closely spaced, summer surfaces. It would be hard |
| 485 | to imagine a depositional behavior that produces such a "grouped peak" pattern at the       |
| 486 | observed regularity. Considering the additional Figure A1, it appears much more plausible   |
| 487 | to assign this to sub-seasonal structure resolved by LA-ICP-MS.                             |
| 488 |                                                                                             |
| 489 | The different age depth relationships displayed in Fig. 5 for KCC and KCI between 150 years |
| 490 | BP and 700-800 years BP needs to be carefully discussed. In fact, within this interval both |
| 491 | cores do not depend on absolute time horizons.                                              |
| 492 | As discussed above, unfortunately there is no means to identify additional horizons in the  |
| 493 | respective time period. We have accounted for the respective uncertainty in annual layer    |
| 494 | counting accordingly. Nonetheless, we have added more text to point out this circumstance.  |
|     |                                                                                             |

495 **Changes to manuscript:** Page 10, Line 32-34: Added a respective statement to the text. 496 497 Caption Fig. 4b The year to year correspondence between Ca annual layers determined 498 by LA and CFA should be indicated drawing lines connecting the star symbols. At the 499 moment no clear one-to-one link is apparent. 500 Although the overall pattern between the CFA signal and the general baseline in LA-ICP-MS 501 is highly similar, we are cautious about necessarily assigning a one-to-one correspondence 502 between peaks. This is not least due to the possibility of a slight remaining offset between 503 the two depth scales. We have added text to fully explain this in the caption. It was our 504 intention to demonstrate that for a given depth interval, the number of years counted in 505 both signals (CFA and LA-ICP-MS) is consistent within uncertainty. 506 **Changes to manuscript:** Added text to the caption of what is now Figure 3. 507 508 P12 L10 "The frequency of occurrence in these total snow loss events is, however, 509 extremely hard to quantify. Counting annual layers in between the above mentioned (dust) 510 horizons within the last century, reveals an offset of typically only one to two years as 511 compared to the known age of the horizons." The last century is not very much 512 representative of colder periods (Little Ice Age, LIA) where snow drift could have been far 513 more important, possibly eroding more annual layers. Notably the LIA is also the time when 514 no absolute time horizons are available for the time scale that depends entirely on counting 515 annual layers. 516 Agreed, but there is probably not much that can be changed about this until additional 517 absolute horizons are discovered. To reiterate, we have employed a dedicated approach to 518 quantify our counting uncertainty. 519 520 P13 L13 "(e.g. note the distinct isotope minima around 1360 AD)." This is an important note 521 when considering the companion paper by More et al. 2017 in Geohealth as this time 522 corresponds almost exactly with the time of the Black Death. Could this isotope minima 523 have been a large winter snow accumulation event? In this sense the implications for the 524 interpretation of the linked Pb record could be very important. 525 We point out that the two papers were not designed to be companion papers, e.g. as stated 526 in the manuscript there are minor differences in the used age scale. Although the 1360 AD 527 isotope minima could be in principle be connected to a higher percentage of winter snow

| 528 | preservation, the signal in the impurity species is less outstanding. We thank the referee for    |
|-----|---------------------------------------------------------------------------------------------------|
| 529 | noting this and take this comment as encouragement for further investigation.                     |
| 530 |                                                                                                   |
| 531 | P14 L14 "However, this is the first time that the correlation holds to this extent also for the   |
| 532 | comparatively old core sections".                                                                 |
| 533 |                                                                                                   |
| 534 | As far as I know, this has been also observed at least in the Ortles ice cores (Gabrielli         |
| 535 | et al. The Cryosphere 10, 2779–2797, 2016).                                                       |
| 536 | This statement refers primarily to the CG ice cores, and we have changed the wording              |
| 537 | accordingly. That said, to our knowledge Gabrielli et al. (2016) show a comparison of the         |
| 538 | three Ortles ice cores on a depth scale, not as time series.                                      |
| 539 | Changes to manuscript: Page 12, Line 17: "However, this is the first time that the                |
| 540 | correlation holds to this extent also for the comparatively old core sections of CG cores"        |
| 541 |                                                                                                   |
| 542 | P15 L5 "Stack of the two stable isotope records (calculated as their simple average)".            |
| 543 |                                                                                                   |
| 544 | Please, mention the temporal step used to calculate averages.                                     |
| 545 | Changed accordingly.                                                                              |
| 546 | Changes to manuscript: Page 13, Line 32: "Stack of the two stable isotope records                 |
| 547 | (calculated as their simple average, at nominal annual resolution)"                               |
| 548 |                                                                                                   |
| 549 | P15 L15 "substantially higher sensitivity values for KCI than KCC, revealing 2.3 vs. 1.4 per      |
| 550 | mill/C, respectively".                                                                            |
| 551 |                                                                                                   |
| 552 | This is surprising considering the striking similarities of the two stable isotope profiles (Fig. |
| 553 | 6). Could you provide an explanation?                                                             |
| 554 | We do not have a full explanation for this phenomenon, adding to the enigmatic nature of          |
| 555 | the isotope sensitivity at CG. However, the central difference between KCI and KCC is the         |
| 556 | especially low net accumulation at KCI. This entails an even stricter confinement to              |
| 557 | sampling mainly the summer season precipitation. This points towards the depositional             |
| 558 | bias to play a role in explaining the high sensitivity values.                                    |
| 559 |                                                                                                   |
| 560 | P15 L16 "changes in snow preservation are expected to bias sensitivity (cf. the conceptual        |

| 561 | consideration in section | 2.2). | " |
|-----|--------------------------|-------|---|
|-----|--------------------------|-------|---|

In this case, the conceptual model presented should be useful to quantify and perhapscorrect this bias.

565 This statement refers to the reasoning of behind the previous comment (see above). While

using the model to correct for the bias is an interesting suggestion, it would certainly

require information on the past variability in snow deposition at each site, which is not

available and generally very hard to quantify at CG. We decided to reword this paragraph toclarify our reasoning.

**570 Changes to manuscript:** Page 14, Line 10ff.: Reworded the paragraph accordingly.

571

572 P15 L17 "This is consistent with the sensitivity difference among KCI and KCC, since an even

573 more strict confinement towards sampling the high summer season can be expected for the 574 lower accumulation KCI."

575

576 Agree, but from just a look of the two records, KCI and KCC show very similar absolute

577 stable isotopes values during the potential calibration time (Fig. 6).

578 The difference is in fact small compared to the absolute values (e.g. for the last 100 years -

579 13.48 vs -13.71 per mil for KCC and KCI, respectively). However, it also affects the

580 magnitude of long term trends, e.g. the recent increase in isotope values over the last 100

581 years (e.g. based on linear regression 0.20 vs. 0.25 per mil / decade for KCC and KCI,

respectively). We thank the reviewer for pointing this out and have included a respectiveremark in the text.

**584 Changes to manuscript:** Page 14, Line 10ff.: Reworded the statement accordingly.

585

586 P17 L3 "While the particle signal alone is not sufficient for differentiating these events,

587 Saharan dust layers in CG ice cores can be reliably identified based on the analyses of Ca2+,

588 supplemented by alkalinity measurements and, in principle, particle size distribution

589 (Wagenbach et al., 1996)."

590

591 This information is not sufficient to discriminate between a Saharan dust event and a past

592 summer surface formed by dry dust accumulation. While Saharan dust events may well

593 have these characteristics, also summer dust layers formed during prolonged dry periods

| 594 | could have the same or similar characteristics. In addition, a higher coarse particle              |
|-----|----------------------------------------------------------------------------------------------------|
| 595 | percentage may be more indicative of local dust rather than long-range transported dust.           |
| 596 | In order to avoid redundancies we would like to refer to our previous responses presented          |
| 597 | above, and references to the earlier studies.                                                      |
| 598 |                                                                                                    |
| 599 | P18 L5 "This is in broad agreement with periods of enhanced Saharan dust deposition                |
| 600 | reported by Thevenon et al. (2009) obtained from elemental analysis in a CG ice core."             |
| 601 |                                                                                                    |
| 602 | This presumed broad agreement should be demonstrated in detail. At this time the high              |
| 603 | frequency of dust events in the described periods is both consistent with more frequent            |
| 604 | Saharan events and the formation of dust enriched past summer surfaces.                            |
| 605 | Accepting the limitations due to the different depth resolution in both records, we made an        |
| 606 | attempt to compare our findings with the results by Thevenon et al. (2009) in a little more        |
| 607 | detail.                                                                                            |
| 608 | Changes to manuscript: Page 17, Lines 7 ff.: Added text.                                           |
| 609 |                                                                                                    |
| 610 | P19 L25 "The connection of the distinct increase in coarse particles with enhanced dust            |
| 611 | event frequency indicates an increase in direct transport of Saharan dust the (as opposed to       |
| 612 | indirect advection with longer pathway and thus stronger decrease in coarse particles)."           |
| 613 |                                                                                                    |
| 614 | This is, at best, consistent (not indicative) with an increase in direct transport of Saharan      |
| 615 | dust. In fact this observation is also compatible with other scenarios (e.g. higher occurrence     |
| 616 | of summer surfaces marked by dust, increase in the intensity of the summer vertical                |
| 617 | transport of dust of Alpine origin).                                                               |
| 618 | Based on our response presented above regarding the distinct difference in particle sizes of       |
| 619 | Saharan dust (Wagenbach et al. 1996, Wagenbach and Geis 1988), in our view this                    |
| 620 | observation is not compatible with the mentioned scenarios. However, we have slightly              |
| 621 | changed the wording of this statement following the referees suggestion.                           |
| 622 | Changes to manuscript: Page 20, Line 11: "the connection of the distinct increase in coarse |
| 623 | particles with enhanced dust event frequency rather suggests an increase in direct          |
| 624 | transport of Saharan dust"                                                                         |
| 625 |                                                                                                    |

626 P19 L27 "Notably, this view is consistent with increased deuterium excess, which would be

| 627 exp | pected from wa | arm and dry air | masses collecting | g moisture over the | e Mediterranean." |
|---------|----------------|-----------------|-------------------|---------------------|-------------------|
|---------|----------------|-----------------|-------------------|---------------------|-------------------|

- 628
- 629 This may be an important note that needs to be expanded and adequately referenced.
- 630 Finding increased values of deuterium excess supports the view of a relative increase in
- 631 Saharan dust advection, which we have made more clear. We have also added an according
- 632 reference as suggested.
- 633 **Changes to manuscript:** Page 20, Line 15ff.: Changed accordingly.
- 634
- 635 P20 L12 "increased meridional transport favoring direct Saharan dust advection over the
- 636 Mediterranean is consistent with a NAO+ dominated MCA, lasting 1100 to about 1300 AD,
- 637 as proposed by Trouet et al. (2009). Although NAO is mainly a winter signal (thus not a
- 638 direct concern to CG summer representative ice core signals)"
- 639
- 640 As mentioned, Saharan dust advection is just one of the possibilities and a NAO signal
- 641 consistent with this hypothesis is really a weak reasoning, especially considering that a NAO
- 642 winter signal can have little impact on the summer biased cores from Colle Gnifetti. Please,
- 643 consider to entirely remove this section (lines 10-17).
- 644 Following the reviewers comment and after additional consideration, we have decided to645 remove this section.
- 646
- 647 Conclusions: conclusions needs to be tuned down accordingly to the main observations
- 648 performed.
- 649 Changed accordingly.
- 650 Changes to manuscript: Page 20, Line 25ff.: Reworded the second part of the conclusions.651
- 652 P21-L12 "The intrinsic contribution of snow preservation may bias the isotope temperature653 sensitivity".
- 055 Selisiti
- 654
- 655 This is interesting but it has not been shown and adequately discussed within the text.
- 656 We have removed this statement from the conclusions.
- 657

| 1  | "Temperature and mineral dust variability recorded in two low accumulation Alpine           |
|----|---------------------------------------------------------------------------------------------|
| 2  | ice cores over the last millennium" by Pascal Bohleber et al.                               |
| 3  | - Response to reviews -                                                                     |
| 4  |                                                                                             |
| 5  | Please note:                                                                                |
| 6  | • All line numbers in "Changes to manuscript" refer to the new version (if not              |
| 7  | noted otherwise)                                                                            |
| 8  | • Changes in the corresponding pdf are highlighted in red                                   |
| 9  | • Author's responses to the referee's comments are in blue                                  |
| 10 | • All new references can be found in the new manuscript                                     |
| 11 |                                                                                             |
| 12 | Introductory remark:                                                                        |
| 13 | We thank both referees for their very thorough reviews and we appreciate the helpful        |
| 14 | suggestions and comments. After careful consideration, especially of points commonly        |
| 15 | raised by both reviewers, we determined the need to clarify our basic line of argument. For |
| 16 | this purpose, we would like to emphasize the following key points:                          |
| 17 | 1) We aim to distinguish throughout the paper two separate signal components of the         |
| 18 | Ca2+ record: 1. Episodic spikes , typically two orders of magnitude above            |
| 19 | background levels, and 2. Long-term trends of the decadal-scale average Ca2+         |
| 20 | concentration. Both components are evaluated separately. At CG, mineral                     |
| 21 | background aerosol levels are generally low and the Ca2+ record is dominated by             |
| 22 | inputs of Saharan dust (e.g. Wagenbach et al. 1996). In this sense, the already             |
| 23 | established link between Ca2+ and Saharan dust concerns signal component 1. The             |
| 24 | potential new link between Ca2+ and temperature is evaluated for signal                     |
| 25 | component 2.                                                                                |
| 26 | 2) Regarding 1., we do not intend to make quantitative inferences regarding mineral         |
| 27 | dust concentrations of individual events but aim to estimate their frequency of             |
| 28 | occurrence at CG. For this purpose we build on what has already been demonstrated           |
| 29 | in previous studies, namely that Ca2+ combined with an alkalinity measure is in fact        |
| 30 | a sensitive and appropriate tool to identify Saharan dust layers at CG (Wagenbach et        |
| 31 | al., 1996).                                                                                 |
| 32 | 3) Regarding 2., we respond to the intriguing present situation at CG where we face i)      |
| 33 | fundamental shortcomings in making quantitative use of the stable water isotope             |

| 34 |         | thermometer (Bohleber et al. 2013) and ii) the already known co-variation between          |
|----|---------|--------------------------------------------------------------------------------------------|
| 35 |         | trends in Ca2+ and delta O-18 (Wagenbach et al. 1996, Wagenbach and Geis, 1989)            |
| 36 |         | as well as delta 0-18 and instrumental temperature (Bohleber et al. 2013). This            |
| 37 |         | raises the question to what extent a relationship exists between temperature and           |
| 38 |         | Ca2+ trends, and if this may serve as a potential substitute for quantitative              |
| 39 |         | temperature reconstruction at CG.                                                          |
| 40 | 4)      | While we explore the suggested relation between Ca2+ trends and temperature, we            |
| 41 |         | strongly emphasize that it is not our intention to introduce a new ice core                |
| 42 |         | temperature proxy. We evaluate the Ca2+ trends solely regarding their site-specific |
| 43 |         | temperature connection. This is an analogue approach as pursued for NH4+ in the            |
| 44 |         | Bolivian Andes (Kellerhals et al. 2010).                                                   |
| 45 | 5)      | We also emphasize that we by no means disregard the influence of snow deposition           |
| 46 |         | and post-depositional effects. In fact, the main goal in using the semi-quantitative       |
| 47 |         | snow deposition model (section 2.2) is to demonstrate that post-depositional               |
| 48 |         | influence may not be disregarded when evaluating the temperature coupling to               |
| 49 |         | Ca2+-trends.                                                                               |
| 50 | In orde | er to eliminate the apparent ambiguities in the original version and in order to make      |
| 51 | our lin | e of argument more clear we have made the following major changes to the                   |
| 52 | manus   | cript. We feel that by means of these changes the most important issues raised by the      |
| 53 | review  | ers have been properly addressed and the clarity of the paper has been substantially       |
| 54 | improv  | red. Detailed responses to the referees' comments are given separately for referee #1      |
| 55 | and #2  | , respectively.                                                                            |
| 56 | Chang   | es to manuscript:                                                                          |
| 57 | •       | We have clarified the abstract and the conclusions according the above points. We          |
| 58 |         | now present two additional tables and two additional figures as supporting                 |
| 59 |         | evidence in the appendix / as supplementary material.                                      |
| 60 | •       | Page 4 Lines 29ff.: We added a clear statement regarding the separate treatment of         |
| 61 |         | Ca2+-spikes and long-term variability in this study.                                       |
| 62 | •       | We have split up the previous section 2.2. as follows:                                     |
| 63 |         | • Page 3 Lines 16ff.: We combined with the original section 2.1 the                        |
| 64 |         | fundamental description of snow preservation at CG and its consequences                    |
| 65 |         | for interpreting the isotope and mineral dust proxies. This also includes the              |

| 66 | basic reasoning for expecting a temperature-related imprint in the long-                         |  |  |  |
|----|--------------------------------------------------------------------------------------------------|--|--|--|
| 67 | term Ca2+-variability.                                                                           |  |  |  |
| 68 | • Since we feel like it has diverted the attention from our main line of                         |  |  |  |
| 69 | argument, we have moved the details of the semi-quantitative treatment of                        |  |  |  |
| 70 | snow deposition to the supplementary material in the appendix.                                   |  |  |  |
| 71 | • Page 18, Line 4ff.: We now refer to the semi-quantitative analysis at a later                  |  |  |  |
| 72 | point in the manuscript. The discussion of potential causes of the observed                      |  |  |  |
| 73 | Ca2+-temperature co-variation is now presented within 5. Results and                             |  |  |  |
| 74 | Discussion. We believe this makes it easier to follow for the reader, since the                  |  |  |  |
| 75 | results have been presented at that point.                                                       |  |  |  |
| 76 | • Page 18, Line 21ff: We have included a clear statement regarding the site-specific             |  |  |  |
| 77 | nature of the observed Ca2+-temperature connection.                                              |  |  |  |
| 78 |                                                                                                  |  |  |  |
| 79 | Response to anonymous referee #2                                                                 |  |  |  |
| 80 | The paper presents an excellent dataset of stable water isotopes and other 'dust' proxies        |  |  |  |
| 81 | (i.e. insoluble particles and Ca2+) from two separate ice cores drilled at Colle Gnifetti in the |  |  |  |
| 82 | Pennine Alps, reaching back in time as far as a thousand year, a remarkable achievement for      |  |  |  |
| 83 | a European alpine ice core. This study combines a very good quality of data retrieval with a     |  |  |  |
| 84 | robust strategy regarding the dating, and therefore deserves to be published in Climate of       |  |  |  |
| 85 | the Past. The data treatment and statistical approach is also adequate and robust and only       |  |  |  |
| 86 | minor changes should be made. I will illustrate now few of the weaknesses that the               |  |  |  |
| 87 | manuscript presents and some suggestions on how to strengthen these points before the            |  |  |  |
| 88 | final publication. Detailed comments follow.                                                     |  |  |  |
| 89 | We thank the referee for the comments and encouragement to further strengthen the                |  |  |  |
| 90 | manuscript.                                                                                      |  |  |  |
| 91 |                                                                                                  |  |  |  |
| 92 | Firstly, the manuscript fails a bit in illustrating the reason why it is important to obtain a   |  |  |  |
| 93 | Ca2+-derived temperature profile and what advantages/disadvantages this would have               |  |  |  |
| 94 | compared to a conventional $\delta 180$ -derived temperature profile. As mentioned in the        |  |  |  |
| 95 | abstract, the high and potentially non-stationary isotope/temperature sensitivity limits the     |  |  |  |
| 96 | quantitative use of the stable isotope ( $\delta 180$ ) variability and therefore a Ca2+-derived |  |  |  |
| 97 | temperature profile could provide essential information for a better constrain of                |  |  |  |
| 98 | temperature variability in the deepest (oldest) section of the two ice cores. This point         |  |  |  |

99 should be highlighted more considering, however, that: i) Ca2+ sensitivity to temperature 100 changes might be, and it is likely to be, non-stationary as well over the last 1000 yrs; ii) the 101 relationship between Ca2+ and temperature could very well derive from post-depositional 102 processes. This last point is particularly relevant (also considering that NH4 show a similar 103 temperature dependance) and the authors should elaborate more on why they think this is 104 not the case. For example, if there is any data available of density, DEP or occurrence of melt 105 layers, I suggest that the authors should use these data to back up some of their assumption 106 regarding the summer-signal preservation by consolidation and its relationship with the 107 seasonality of Ca2+.

108 We thank the referee for this comment, in particular for the suggestion to include

109 considering the density profile of the core. As discussed in the initial remarks, considering

110 the reviews we realized that a few issues need to be clarified, and see some of these points

arising here, too. In fact, we believe that post-depositional processes must be considered

112 when explaining the apparent coupling between temperature and long-term variability of

113 Ca2+. We have clarified and extended our discussion of this point. The comparison between

density and Ca2+ data clearly shows that dust-rich layers are coinciding with locally

115 enhanced density, that stem from fast snow consolidation. This "self-preserving"

116 characteristic of Ca2+ (and other dust-related species) against wind erosion is one of the

117 main differences with respect to the stable isotope signal. We have also added text to

118 discuss the fact that, while a non-stationary character of the Ca2+-temperature relationship

119 is certainly a possibility, we find no evidence for this within the instrumental period (in

120 contrast to the stable isotopes). Following the referees comment we have also elaborated

121 that, for this reason, fundamental shortcoming exists in quantitatively interpreting the

122 isotope-thermometer over long time scales at CG. Although we do not intend to introduce

123 Ca2+ as a new general temperature indicator, we see our findings as a strong indication of

124 the potential for using the long-term variability of Ca2+ as a *site-specific* temperature proxy.

125 We have clarified this view also in our conclusions.

126 **Changes to manuscript**:

Page 4, Lines 4ff.: Rewrote part of this section accordingly. Specifically regarding the
 motivation for expecting a temperature-related imprint in Ca2+.

Page 18, Lines 4ff.: Moved and rewrote part of the paragraph (originally in section
2.2), specifically mentioning the self-preserving character of Ca2+.

• Page 13, Lines 14ff: Included additional mentioning of the shortcomings of the

- 132 stable isotope thermometer at CG.
- Page 19, Lines 13ff.: Included a statement to clarify the lack of evidence for a non stationary Ca2+-temperature relationship.
- Page 18, Lines 21ff: Emphasized the site-specific role of the Ca2+-temperature
  association.
- 137

138Furthermore, the assumption that the Ca2+ signal is almost entirely expression of a dust

- 139 input from Saharan region is not enough justified in the text. The fact that the Ca2+ profile
- 140 might derive from both wet and dry deposition and both proximal and distal sources cannot
- 141 be ruled out from the data shown in the manuscript. Since the isotope/impurity co-
- 142 variation on the inter-annual scale is mainly related to changes in the amount of winter
- 143 precipitation contributing to annual mean values, I think is necessary to briefly consider
- 144 different scenarios concerning the (although marginal) role of dry deposition in the Colle
- 145 Gnifetti area and how these could change the Ca2+ signal in the different cases.
- 146 We would like to refer here to the initial comments and point out that at CG, mineral
- background aerosol levels are generally low (including summer) and the Ca2+ record is
- 148 dominated by inputs of Saharan dust, which has been demonstrated in previous studies
- 149 (Wagenbach et al. 1996). Thank you also for pointing out the role of dry deposition, which
- 150 we have so far not explicitly mentioned in the manuscript.
- **151 Changes to manuscript:** Page 4, Lines 13ff.: We have included a brief discussion of the
- 152 contribution made by dry deposition to the mineral dust content at CG.
- 153
- 154 While provenance studies (Sr and Nd isotopes for example) go beyond the scope of the
- 155 work, I think a more detailed discussion on the comparison of the insoluble dust profile vs
- the Ca2+ profile is necessary to utilize the calcium signal a proxy for Saharan dust input.
- 157 We agree with the referee that a provenance study based on isotopic trace element analysis
- 158 exceeds the scope of this study. At the same time the identification of Saharan dust input
- based on Ca2+ (and an alkalinity measure) has already been established in a previous study
- 160 (Wagenbach et al. 1996). Thus we did not intend to develop a new (and arguably more
- 161 precise) proxy for Saharan dust events at CG, but intended to use this already established
- tool. We have clarified this in the respective introductory section 2.
- 163 **Changes to manuscript:** Page 4, Lines 26ff.: Added a clarifying statement regarding the
- 164 tool to identify Saharan dust events.

166 Whether Saharan dust-Ca2+ data is a reliable proxy for palaeotemperature is yet again 167 another point that needs to be better illustrated in the text. I think the authors should 168 provide more justification regarding why the Ca2+ variability is mainly related to 169 temperature changes and not, for instance, to changes at the dust source (Saharan desert). 170 As outlined above it is not our intention to directly link the Saharan-dust component of the 171 Ca2+ data (spikes) to temperature, but rather investigate for this purpose the long-term 172 variability of Ca2+. We find the Ca2+ trends in surprisingly good correlation with 173 instrumental temperature throughout the full instrumental period, and go on to discuss 174 how snow preservation plays a decisive role in introducing this Ca2+-temperature coupling. 175 It seems likely that only large and systematic changes at the dust source would change the 176 long-term Ca2+ variability, or eventually override the coupling to temperature. On the other 177 hand, these changes (e.g. increased dust mobilization) would likely also influence the 178 Saharan dust spikes and their frequency of occurrence. However, the only instance where 179 we find an outstanding according feature is the increased dust occurrence in the medieval 180 period of our record. We thank the referee for this suggestion and now consider this issue 181 in our discussion. 182 Changes to manuscript: Page 19, Line 18: Added text to discuss the role of changes at the 183 dust source. 184 185 Detailed comments: 186 Page 1 Line 1-2: I would update this statement in view of the recent 7000-yrs long ice core 187 record from the Ortles (Gabrielli et al., 2017). 188 Changed accordingly to clarify. In contrast to Ortles, Colle Gnifetti is a non-temperate site. 189 190 Page 3 Line 11-12: "which prevents any link of the climatologic precipitation rate to the net 191 snow accumulation rate". I am not sure I understand here: Does this mean that the 192 seasonality in the proxies is not governed by accumulation rate? Or is rather the longer-193 time variability? In any case I suggest changing the word "prevents" with "limits". 194 What we intend to say is that due to the highly variably snow deposition at CG, it is not 195 possible to infer precipitation changes based on e.g. annual layer thickness (e.g. as done 196 with Greenland ice cores). We have clarified the wording accordingly. 197 Changes to manuscript: Page 3, Line 10-11: "limits linking the net snow accumulation rate

| 198 | to the climatologic precipitation rate"                                                       |
|-----|-----------------------------------------------------------------------------------------------|
| 199 |                                                                                               |
| 200 | Page 3 Line 17: I found the wording a bit confusing. What "chemical/isotopic conditions"      |
| 201 | means? Do you mean chemical and isotopic signatures?                                          |
| 202 | Yes we mean the signature of chemical and isotopic species measured in the CG ice cores.      |
| 203 | We have clarified the wording accordingly.                                                    |
| 204 | Changes to manuscript: Page 3, Line 16: "chemical and isotopic signatures "                   |
| 205 |                                                                                               |
| 206 | Page 4 Line 1-2: "the isotope/impurity co-variation on the inter-annual scale reflects to a   |
| 207 | large degree changes in the amount of winter precipitation contributing to annual mean        |
| 208 | values" I think is important here to highlight why the authors think dry deposition is        |
| 209 | playing a marginal role.                                                                      |
| 210 | See our response above, we now include a short discussion of the role of dry deposition.      |
| 211 | Changes to manuscript: Page 4, Line 14ff.: Added text regarding dry deposition.               |
| 212 |                                                                                               |
| 213 | Page 4 line 10-11:"Therefore, the Ca2+ record of the CG ice cores is primarily related to     |
| 214 | mineral dust and dominated by Saharan dust". It's hard to tell without provenance studies. I  |
| 215 | suggest using "dominated by dust, most likely originating in the Saharan desert".             |
| 216 | Thank you, we have reworded this statement and included the respective reference.             |
| 217 | Changes to manuscript: Page 4, Line 13-14: Reworded the previous statement.                   |
| 218 |                                                                                               |
| 219 | Page 7 Line 3: "Deviations from a CPP of 50% indicate higher or lower contribution of large   |
| 220 | and small particles respectively". You have to exclude local sources of dust then if you want |
| 221 | to use the threshold to distinguish Saharan dust layers. I would add a sentence justifying    |
| 222 | this.                                                                                         |
| 223 | Thank you. We now point out the findings of Wagenbach and Geis (1988) in this context,        |
| 224 | who showed that Saharan dust in fact differs in volume size distribution in comparison to     |
| 225 | local and background sources.                                                                 |
| 226 | Changes to manuscript: Page 5, Line 16: " The threshold was chosen such that it               |
| 227 | corresponds to the expected median particle diameter of Saharan dust particles at CG,         |
| 228 | which was shown to be distinguishable from background sources"                                |
|     |                                                                                               |

| 230 | Page 8 Line 1: I would specify what "Ca signal" means. Is it Intensity in counts per second?    |
|-----|-------------------------------------------------------------------------------------------------|
| 231 | Or total counts? Please add this also to the relevant figures.                                  |
| 232 | Thank you for pointing this out- in this case it is in fact intensity in counts per second,     |
| 233 | although it is possible to achieve an according calibration of the LA-ICP-MS signal (Sneed et   |
| 234 | al. 2015).                                                                                      |
| 235 | Changes to manuscript: Added text to captions of Figures 2 and 3, respectively.                 |
| 236 |                                                                                                 |
| 237 | Page 9 Line 8: "Below 26 m WE the identification of annual layers became ambiguous and          |
| 238 | was abandoned". Maybe I missed this information, but why then LA-ICPMS was not                  |
| 239 | performed on the KCI core? Please provide justification, if it is not provided somewhere        |
| 240 | else.                                                                                           |
| 241 | There was actually a pilot study for LA-ICP-MS performed on KCI (Sneed et al. 2015),            |
| 242 | however, not targeting yet the identification and counting of annual layers. Given the          |
| 243 | sophisticated and time-consuming nature of LA-ICP-MS we have so far only analysed KCC in        |
| 244 | a continuous manner. We take the comment as encouragement to further pursue the LA-             |
| 245 | ICP-MS analysis, potentially revisiting KCI in the future. We have added text to provide this   |
| 246 | information.                                                                                    |
| 247 | Changes to manuscript: Page 8, Lines 10ff.: Added text regarding LA-ICP-MS on KCI.              |
| 248 |                                                                                                 |
| 249 | Page 13 Line 12: "due to the strong effect of isotope diffusion at CG, inter-annual or even     |
| 250 | seasonal isotope variability is effectively eliminated". What about Ca2+ diffusion? While       |
| 251 | dust does not diffuse, the contribution of soluble particles to the Ca44 signal should be       |
| 252 | briefly addressed too, together with their possible diffusion.                                  |
| 253 | The effect of diffusion is certainly smaller for Ca2+ than for the stable water isotopes, as we |
| 254 | do not see any evidence of diffusion hampering the identification of the annual layers at the   |
| 255 | high resolution afforded by LA-ICP-MS. However, we are now mentioning this effect, and in       |
| 256 | particular also point out the contribution of soluble Ca to the LA-ICP-MS signal.               |
| 257 | Changes to manuscript:                                                                          |
| 258 | • Page 6, Line 13: "The 44Ca signal comprises contributions of soluble and insoluble            |
| 259 | Ca"                                                                                             |
| 260 | • Page 9, Line 16-17: "The annual layer signal remains clearly identifiable for the             |
| 261 | remaining part of the depth-range investigated here (e.g. apparently not affected by            |
| 262 | diffusion of soluble Ca)"                                                                       |

| 264 | Page 14 Line 31-32: "From a preliminary inspection of snow pit data recently obtained for             |
|-----|-------------------------------------------------------------------------------------------------------|
| 265 | the KCI-KCC flow line, there is no clear indication of a systematic trend in mean $\delta 180$ levels |
| 266 | upstream of KCC, however." It might be worthy to consider adding a plot (at least in the              |
| 267 | supplementary material) showing this.                                                                 |
| 268 | As mentioned in the text the detailed investigation of the isotope-upstream effect is still           |
| 269 | ongoing based on sophisticated 3D-flow modelling (PhD thesis Carlo Licciulli at Heidelberg            |
| 270 | University). However, we have provided additional information regarding the preliminary               |
| 271 | inspection.                                                                                           |
| 272 | Changes to manuscript: Page 13, Lines 24ff.: Added text accordingly.                                  |
| 273 |                                                                                                       |
| 274 | Page 15 Line 12: "higher sensitivity values for KCI than KCC, revealing 2.3 vs. 1.4 $\%$ /°C,         |
| 275 | respectively".                                                                                        |
| 276 | This discrepancy seems surprisingly high even considering the difference in accumulation              |
| 277 | rate that you correctly highlight. Could it be related also to the strong isotope diffusion at        |
| 278 | CG?                                                                                                   |
| 279 | The degree of isotope diffusion could certainly be another difference between KCI and KCC,            |
| 280 | thank you for pointing this out. This is especially so in the firn section, and here (due to the      |
| 281 | difference in accumulation rate) the age interval represented by the firn column differs for          |
| 282 | the two cores. Although it is difficult at this stage to give a more quantitative evaluation          |
| 283 | regarding its effect on isotope sensitivity, we now include mention isotope diffusion. We             |
| 284 | will also consider this in a potential future investigation on the enhanced isotope sensitivity.      |
| 285 | Changes to manuscript: Page 14, Line 13-14: Added text accordingly.                                   |
| 286 |                                                                                                       |
| 287 | Page 20 Line 10-17: This entire section seems a bit far-fetched. As the authors said, the             |
| 288 | summer-bias signal at CG strongly advocate against a NAO imprint on the KCC and KCI                   |
| 289 | temperature reconstruction. I suggest adding few more considerations to justify this link or          |
| 290 | remove the entire section.                                                                            |
| 291 | After considering the comments of both referees in this direction, we decided to remove               |
| 292 | this section from the discussion.                                                                     |
|     |                                                                                                       |

- Page 21 Line 1-20: I suggest to the authors to add a sentence outlining the feasibility of
- using Ca2+ records for temperature reconstruction in other alpine site, or generally in other
- low accumulation ice core site.
- 297 We are now generally trying to be more clear about the site-specific nature of the potential
- temperature significance of the Ca2+ long-term variability. However, it would be interesting
- to test if the Ca2+-temperature association observed at CG holds also at other alpine sites.
- **300 Changes to manuscript:** Page 20, Line 28: Included a respective statement in the
- 301 conclusions.
- 302
- 303 References
- 304 Gabrielli, P., Barbante, C., Bertagna, G., Bertó, M., Carturan, L., Dinale, R., & Seppi, R. (2017,
- April). 7000 year European climate record from the Ortles ice core. In EGU General
- 306 Assembly Conference Abstracts (Vol. 19, p. 9932).
- 307

[revised manuscript text omitted]
 14C content. Calibration of the retrieved 14C ages was performed using OxCal version 2.4

---

## Author Response (AR2)

**"Temperature and mineral dust variability recorded in two low accumulation Alpine**
**ice cores over the last millennium" by Pascal Bohleber et al.**
- final technical corrections -
*Please note:*
- *All line numbers in "Changes to manuscript" refer to the new version (if not*
*noted otherwise)*
- *Changes in the corresponding pdf are highlighted in red*
- *Author's responses to the referee's comments are in blue*
- *All new references can be found in the new manuscript*
*1. Comments by the Editor*
Comments to the Author:
The comments of the two referees have been carefully addressed and especially, it is now
very clear how the Ca2+ vs temperature link should be considered.
I would still suggest a small change in the conclusion section: l. 30, p. 20, can you be more
cautious about the link between Ca2+ and temperature and change the sentence beginning
with "Eploiting ..." ? I suggest to remove the first part of the sentence and simply
mentionning "Considering a constant Ca2+ - temperature relationship (1) proves ...."
Many thanks again for your submission and careful revisions.
Thank you again for your helpful comments. We appreciate the suggestion and have
changed the sentence accordingly. Please also note the following additional technical
changes to the manuscript:
- We have updated and added the following references: Hoffmann et al. 2017b,
Spaulding et al. 2017, Luongo et al. 2017
- Updated Acknowledgements
- We have included the former appendix as a separate document serving as
supplementary material. We have changed the references in the text accordingly.
- We have submitted the central datasets of this study to the Pangaea repository and
will add a respective reference at the end of the manuscript as soon as they are
available.

*2. Comments by the Anonymous Referee #1*

I find this version of the paper considerably improved and I believe the authors clarified my most important points. However, the authors may consider these additional comments.

We thank the referee again for the constructive comments and thorough review. We have also considered the additional remarks for the final version of the manuscript (see below).

Main points:

I think the supplementary figures A1 and A2 provide important additional information/value to the paper.

Figure A1: This figure provides a good idea of 1) how the seasonal cycles of delta 18O and

Ca2+ can be preserved at Colle Gnifetti and 2) the partitioning of summer and winter accumulation. While I see how the large amplitude seasonal signal can be conserved and transferred to greater depths over time, I do not see how the same can happen for the minor

Ca2+ peaks, as indicated by the arrows. To me this figure does not support the concept of

"group of peaks" observed at greater depths and referred as annual cycle. Counting these groups as annual layers is still equivocal and, while this could still be correct, (another unknown post-depositional glaciological mechanism may concur to form this presumed annual feature) the dating of this core remain well constrained only in its upper and bottom parts. Identifying absolute time horizons in the time interval of 100-300 years before present, seems however a frequent problem of Alpine cores whose dating might typically suffer of a lower relative accuracy at that time.

Figure A2: This figure strongly supports the idea that the occurrence of dust in snow layers facilitates their consolidation and thus preservation on site. I was wondering whether it would be possible to also display and comment delta 18O in this figure, as it might be telling about the role of atmospheric temperature and the timing of occurrence of this consolidation process. In any event I strongly recommend this figure to be incorporated within the main text as this seems the most convincing evidence supporting the arguments proposed by the authors.

We appreciate the referee's statement that the new supplementary figures add to the value and clarity of the paper. We have moved figure A2 to the main text as suggested.

The indication of the potential source area of dust remains elusive to me. However, regardless whether the Sahara is constantly the source of these large inputs of dust to Colle

Gnifetti, this does not influence the main message of the paper. I would only recommend the authors to provide some significant references for isotopic and elemental tools/methods (e.g. Sr and Nd isotopes, REE etc.) that could possibly be employed in the future to better constrain potential source areas of dust that reached Colle Gnifetti.

We have added references accordingly. Page 20, Lines 16-17.

A graphical comparison with the Thevenon record might be also helpful.

Since we now discuss the comparison with the Thevenon et al. (2009) dataset in more detail in the text, we have decided not to add another figure in the manuscript.

I would also recommend a final polishing of the English as it does not always sound correct to me.

We will wait for the English language copy-editing to the final stage of manuscript production.

Specific points:

Pag 1, Line 12: I would suggest to replace "advection" with "convection".

In this case we specifically want to include the (comparatively long range) advection of dust and hence leave the wording unchanged.

Pag 3, Line 17-18 "A changing amount of winter precipitation contributing to annual mean values may introduce a coupling on the inter-annual scale among seasonal varying signals, including $\delta18O$ and most impurities". This sentence is not clear.

We have reworded the sentence to make it more clear. Page 3, Lines 16-18.

Pag. 4, Line 27 I would remove "specific".

Changed accordingly.

Pag. 18 Line 11 (and elsewhere within the text, e.g. in caption of figure A2). You may replace

"snow preservation" with "snow consolidation". This latter is more focused on the process.

Changed accordingly where appropriate.

Pag 18 Line 18 "This revealed that the resulting bias from incomplete snow preservation on the average Ca2+ level is already in the same order of magnitude as the long-term Ca2+

trends found in previous studies". This sentence is not clear.

We have reworded the sentence to make it more clear. Page 18, Lines 18-20.

Pag 20 Line 8 "depend on"

Changed accordingly.

*2. Comments by the Anonymous Referee #2*

The authors have improved the paper and answered many of the points I raised in my original review. I am particularly pleased that the authors provided a more detailed assessment of the influence of snow deposition and post-depositional processes in explaining the apparent Ca2+ - temperature covariation. The discussion section is now clearer and benefits significantly by the introduction of subsection 5.2.2, which addresses the main scientific issues concerning the relation of long term variability of Ca2+ and temperature in a more exhaustive way compared to the original version. I therefore think that the manuscript should be published as it is.

Thank you very much again for your constructive review and comments.

I only suggest the following rewording:

- Fig. 1: "[...] hence providing the same upstream catchment area [...]" rather than "hence providing their same upstream catchment area".

Changed accordingly.

[revised manuscript text omitted]